# ESCAPING SADDLE POINTS FASTER WITH STOCHASTIC MOMENTUM

**Jun-Kun Wang, Chi-Heng Lin, & Jacob Abernethy**
Georgia Institute of Technology
{jimwang,cl3385,prof}@gatech.edu

## ABSTRACT

Stochastic gradient descent (SGD) with stochastic momentum is popular in non-convex stochastic optimization and particularly for the training of deep neural networks. In standard SGD, parameters are updated by improving along the path of the gradient at the current iterate on a batch of examples, where the addition of a "momentum" term biases the update in the direction of the previous change in parameters. In non-stochastic convex optimization one can show that a momentum adjustment provably reduces convergence time in many settings, yet such results have been elusive in the stochastic and non-convex settings. At the same time, a widely-observed empirical phenomenon is that in training deep networks stochastic momentum appears to significantly improve convergence time, variants of it have flourished in the development of other popular update methods, e.g. ADAM (Kingma & Ba (2015)), AMSGrad (Reddi et al. (2018b)), etc. Yet theoretical justification for the use of stochastic momentum has remained a significant open question. In this paper we propose an answer: stochastic momentum improves deep network training because it modifies SGD to escape saddle points faster and, consequently, to more quickly find a second order stationary point. Our theoretical results also shed light on the related question of how to choose the ideal momentum parameter–our analysis suggests that $\beta \in [0, 1)$ should be large (close to 1), which comports with empirical findings. We also provide experimental findings that further validate these conclusions.

## 1 INTRODUCTION

SGD with stochastic momentum has been a de facto algorithm in nonconvex optimization and deep learning. It has been widely adopted for training machine learning models in various applications. Modern techniques in computer vision (e.g.Krizhevsky et al. (2012); He et al. (2016); Cubuk et al. (2018); Gastaldi (2017)), speech recognition (e.g. Amodei et al. (2016)), natural language processing (e.g. Vaswani et al. (2017)), and reinforcement learning (e.g. Silver et al. (2017)) use SGD with stochastic momentum to train models. The advantage of SGD with stochastic momentum has been widely observed (Hoffer et al. (2017); Loshchilov & Hutter (2019); Wilson et al. (2017)). Sutskever et al. (2013) demonstrate that training deep neural nets by SGD with stochastic momentum helps achieving in faster convergence compared with the standard SGD (i.e. without momentum). The success of momentum makes it a necessary tool for designing new optimization algorithms in optimization and deep learning. For example, all the popular variants of adaptive stochastic gradient methods like Adam (Kingma & Ba (2015)) or AMSGrad (Reddi et al. (2018b)) include the use of momentum.

Despite the wide use of stochastic momentum (Algorithm 1) in practice, [1] justification for the clear empirical improvements has remained elusive, as has any mathematical guidelines for actually setting the momentum parameter—it has been observed that large values (e.g. $\beta = 0.9$) work well in practice. It should be noted that Algorithm 1 is the default momentum-method in popular software

---

[1] Heavy ball momentum is the default choice of momentum method in PyTorch and Tensorflow, instead of Nesterov's momentum. See the manual pages https://pytorch.org/docs/stable/_modules/torch/optim/sgd.html and https://www.tensorflow.org/api_docs/python/tf/keras/optimizers/SGD.

---

**Algorithm 1:** SGD with stochastic heavy ball momentum

---
1: Required: Step size parameter $\eta$ and momentum parameter $\beta$.
2: Init: $w_0 \in \mathbb{R}^d$ and $m_{-1} = 0 \in \mathbb{R}^d$.
3: **for** $t = 0$ to $T$ **do**
4:      Given current iterate $w_t$, obtain stochastic gradient $g_t := \nabla f(w_t; \xi_t)$.
5:      Update stochastic momentum $m_t := \beta m_{t-1} + g_t$.
6:      Update iterate $w_{t+1} := w_t - \eta m_t$.
7: **end for**

---

packages such as PyTorch and Tensorflow. In this paper we provide a theoretical analysis for SGD with momentum. We identify some mild conditions that guarantees SGD with stochastic momentum will provably escape saddle points faster than the standard SGD, which provides clear evidence for the benefit of using stochastic momentum. For stochastic heavy ball momentum, a weighted average of stochastic gradients at the visited points is maintained. The new update is computed as the current update minus a step in the direction of the momentum. Our analysis shows that these updates can amplify a component in an escape direction of the saddle points.

In this paper, we focus on finding a second-order stationary point for smooth non-convex optimization by SGD with stochastic heavy ball momentum. Specifically, we consider the stochastic nonconvex optimization problem, $\min_{w \in \mathbb{R}^d} f(w) := \mathbb{E}_{\xi \sim \mathcal{D}}[f(w; \xi)]$, where we overload the notation so that $f(w; \xi)$ represents a stochastic function induced by the randomness $\xi$ while $f(w)$ is the expectation of the stochastic functions. An $(\epsilon, \epsilon)$-second-order stationary point $w$ satisfies

$$\|\nabla f(w)\| \leq \epsilon \text{ and } \nabla^2 f(w) \succeq -\epsilon I. \tag{1}$$

Obtaining a second order guarantee has emerged as a desired goal in the nonconvex optimization community. Since finding a global minimum or even a local minimum in general nonconvex optimization can be NP hard (Anandkumar & Ge (2016); Nie (2015); Murty & Kabadi (1987); Nesterov (2000)), most of the papers in nonconvex optimization target at reaching an approximate second-order stationary point with additional assumptions like Lipschitzness in the gradients and the Hessian (e.g. Allen-Zhu & Li (2018); Carmon & Duchi (2018); Curtis et al. (2017); Daneshmand et al. (2018); Du et al. (2017); Fang et al. (2018; 2019); Ge et al. (2015); Jin et al. (2017; 2019); Kohler & Lucchi (2017); Lei et al. (2017); Lee et al. (2019); Levy (2016); Mokhtari et al. (2018); Nesterov & Polyak (2006); Reddi et al. (2018a); Staib et al. (2019); Tripuraneni et al. (2018); Xu et al. (2018)). [2] We follow these related works for the goal and aim at showing the benefit of the use of the momentum in reaching an $(\epsilon, \epsilon)$-second-order stationary point.

We introduce a required condition, akin to a model assumption made in (Daneshmand et al. (2018)), that ensures the dynamic procedure in Algorithm 2 produces updates with suitable correlation with the negative curvature directions of the function $f$.

**Definition 1.** *Assume, at some time $t$, that the Hessian $H_t = \nabla^2 f(w_t)$ has some eigenvalue smaller than $-\epsilon$ and $\|\nabla f(w_t)\| \leq \epsilon$. Let $v_t$ be the eigenvector corresponding to the smallest eigenvalue of $\nabla^2 f(w_t)$. The stochastic momentum $m_t$ satisfies **Correlated Negative Curvature (CNC)** at $t$ with parameter $\gamma > 0$ if*

$$\mathbb{E}_t[\langle m_t, v_t \rangle^2] \geq \gamma. \tag{2}$$

As we will show, the recursive dynamics of SGD with heavy ball momentum helps in amplifying the escape signal $\gamma$, which allows it to escape saddle points faster.

**Contribution:** We show that, under CNC assumption and some minor constraints that upper-bound parameter $\beta$, if SGD with momentum has properties called *Almost Positively Aligned with Gradient* (APAG), *Almost Positively Correlated with Gradient* (APCG), and *Gradient Alignment or Curvature Exploitation* (GrACE), defined in the later section, then it takes $T = O((1-\beta) \log(1/(1-\beta)\epsilon)\epsilon^{-10})$ iterations to return an $(\epsilon, \epsilon)$ second order stationary point. Alternatively, one can obtain an $(\epsilon, \sqrt{\epsilon})$ second order stationary point in $T = O((1-\beta) \log(1/(1-\beta)\epsilon)\epsilon^{-5})$ iterations. Our theoretical result demonstrates that a larger momentum parameter $\beta$ can help in escaping saddle points faster. As saddle points are pervasive in the loss landscape of optimization and deep learning (Dauphin et al.

---

[2]We apologize that the list is far from exhaustive.

(2014); Choromanska et al. (2015)), the result sheds light on explaining why SGD with momentum enables training faster in optimization and deep learning.

**Notation:** In this paper we use $\mathbb{E}_t[\cdot]$ to represent conditional expectation $\mathbb{E}[\cdot|w_1, w_2, \ldots, w_t]$, which is about fixing the randomness upto but not including $t$ and notice that $w_t$ was determined at $t-1$.

## 2 BACKGROUND

### 2.1 A THOUGHT EXPERIMENT.

Let us provide some high-level intuition about the benefit of stochastic momentum with respect to escaping saddle points. In an iterative update scheme, at some time $t_0$ the parameters $w_{t_0}$ can enter a *saddle point region*, that is a place where Hessian $\nabla^2 f(w_{t_0})$ has a non-trivial negative eigenvalue, say $\lambda_{\min}(\nabla^2 f(w_{t_0})) \leq -\epsilon$, and the gradient $\nabla f(w_{t_0})$ is small in norm, say $\|\nabla f(w_{t_0})\| \leq \epsilon$. The challenge here is that gradient updates may drift only very slowly away from the saddle point, and may not escape this region; see (Du et al. (2017); Lee et al. (2019)) for additional details. On the other hand, if the iterates were to move in one particular direction, namely along $v_{t_0}$ the direction of the smallest eigenvector of $\nabla^2 f(w_{t_0})$, then a fast escape is guaranteed under certain constraints on the

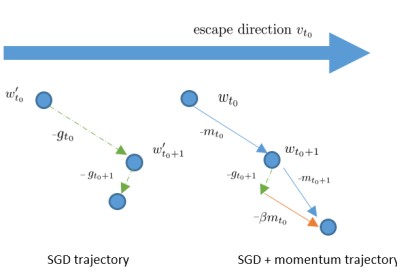

Figure 1: The trajectory of the standard SGD (left) and SGD with momentum (right).

step size $\eta$; see e.g. (Carmon et al. (2018)). While the negative eigenvector could be computed directly, this 2nd-order method is prohibitively expensive and hence we typically aim to rely on gradient methods. With this in mind, Daneshmand et al. (2018), who study non-momentum SGD, make an assumption akin to our CNC property described above that each stochastic gradient $g_{t_0}$ is strongly non-orthogonal to $v_{t_0}$ the direction of large negative curvature. This suffices to drive the updates out of the saddle point region.

In the present paper we study stochastic momentum, and our CNC property requires that the update direction $m_{t_0}$ is strongly non-orthogonal to $v_{t_0}$; more precisely, $\mathbb{E}_{t_0}[\langle m_{t_0}, v_{t_0} \rangle^2] \geq \gamma > 0$. We are able to take advantage of the analysis of (Daneshmand et al. (2018)) to establish that updates begin to escape a saddle point region for similar reasons. Further, this effect is *amplified* in successive iterations through the momentum update when $\beta$ is close to 1. Assume that at some $w_{t_0}$ we have $m_{t_0}$ which possesses significant correlation with the negative curvature direction $v_{t_0}$, then on successive rounds $m_{t_0+1}$ is quite close to $\beta m_{t_0}$, $m_{t_0+2}$ is quite close to $\beta^2 m_{t_0}$, and so forth; see Figure 1 for an example. This provides an intuitive perspective on how momentum might help accelerate the escape process. Yet one might ask *does this procedure provably contribute to the escape process* and, if so, *what is the aggregate performance improvement of the momentum?* We answer the first question in the affirmative, and we answer the second question essentially by showing that momentum can help speed up saddle-point escape by a multiplicative factor of $1 - \beta$. On the negative side, we also show that $\beta$ is constrained and may not be chosen arbitrarily close to 1.

### 2.2 MOMENTUM HELPS ESCAPE SADDLE POINTS: AN EMPIRICAL VIEW

Let us now establish, empirically, the clear benefit of stochastic momentum on the problem of saddle-point escape. We construct two stochastic optimization tasks, and each exhibits at least one significant saddle point. The two objectives are as follows.

$$\min_w f(w) \quad := \quad \frac{1}{n}\sum_{i=1}^n \left(\frac{1}{2}w^\top H w + b_i^\top w + \|w\|_{10}^{10}\right), \tag{3}$$

$$\min_w f(w) \quad := \quad \frac{1}{n}\sum_{i=1}^n \left((a_i^\top w)^2 - y\right)^2. \tag{4}$$

Problem (3) of these was considered by (Staib et al. (2019); Reddi et al. (2018a)) and represents a very straightforward non-convex optimization challenge, with an embedded saddle given by the matrix

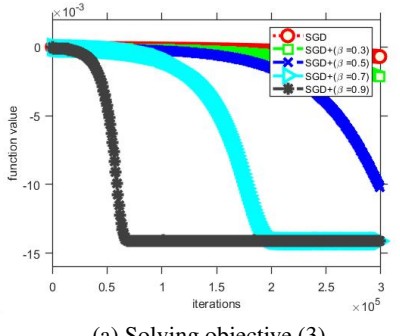 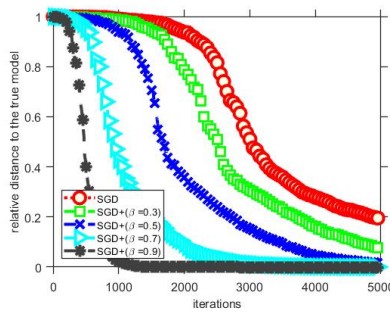

(a) Solving objective (3).  (b) Solving objective (4). (phase retrieval)

Figure 2: Performance of SGD with different values of $\beta = \{0, 0.3, 0.5, 0.7, 0.9\}$; $\beta = 0$ corresponds to the standard SGD. Fig. 4a: We plot convergence in function value $f(\cdot)$ given in (3). Initialization is always set as $w_0 = \mathbf{0}$. All the algorithms use the same step size $\eta = 5 \times 10^{-5}$. Fig. 4b: We plot convergence in *relative distance* to the true model $w^*$, defined as $\min(\|w_t - w^*\|, \|w_t + w^*\|)/\|w^*\|$, which more appropriately captures progress as the global sign of the objective (4) is unrecoverable. All the algorithms are initialized at the same point $w_0 \sim \mathcal{N}(0, \mathcal{I}_d/(10000d))$ and use the same step size $\eta = 5 \times 10^{-4}$.

$H := \text{diag}([1, -0.1])$, and stochastic gaussian perturbations given by $b_i \sim \mathcal{N}(0, \text{diag}([0.1, 0.001]))$; the small variance in the second component provides lower noise in the escape direction. Here we have set $n = 10$. Observe that the origin is in the neighborhood of saddle points and has objective value zero. SGD and SGD with momentum are initialized at the origin in the experiment so that they have to escape saddle points before the convergence. The second objective (4) appears in the *phase retrieval* problem, that has real applications in physical sciences (Candés et al. (2013); Shechtman et al. (2015)). In phase retrieval[3], one wants to find an unknown $w^* \in \mathbb{R}^d$ with access to but a few samples $y_i = (a_i^\top w^*)^2$; the design vector $a_i$ is known a priori. Here we have sampled $w^* \sim \mathcal{N}(0, \mathcal{I}_d/d)$ and $a_i \sim \mathcal{N}(0, \mathcal{I}_d)$ with $d = 10$ and $n = 200$.

The empirical findings, displayed in Figure 2, are quite stark: for both objectives, convergence is significantly accelerated by larger choices of $\beta$. In the first objective (Figure 4a), we see each optimization trajectory entering a saddle point region, apparent from the "flat" progress, yet we observe that large-momentum trajectories escape the saddle much more quickly than those with smaller momentum. A similar affect appears in Figure 4b. To the best of our knowledge, this is the first reported empirical finding that establishes the dramatic speed up of stochastic momentum for finding an optimal solution in phase retrieval.

## 2.3 RELATED WORKS.

**Heavy ball method:** The heavy ball method was originally proposed by Polyak (1964). It has been observed that this algorithm, even in the deterministic setting, provides no convergence speedup over standard gradient descent, except in some highly structure cases such as convex quadratic objectives where an "accelerated" rate is possible (Lessard et al. (2016); Goh (2017); Ghadimi et al. (2015); Sun et al. (2019); Loizou & Richtárik (2017; 2018); Gadat et al. (2016); Yang et al. (2018); Kidambi et al. (2018); Can et al. (2019)). We provide a comprehensive survey of the related works about heavy ball method in Appendix A.

**Reaching a second order stationary point:** As we mentioned earlier, there are many works aim at reaching a second order stationary point. We classify them into two categories: specialized algorithms and simple GD/SGD variants. Specialized algorithms are those designed to exploit the negative curvature explicitly and escape saddle points faster than the ones without the explicit exploitation (e.g. Carmon et al. (2018); Agarwal et al. (2017); Allen-Zhu & Li (2018); Xu et al. (2018)). Simple GD/SGD variants are those with minimal tweaks of standard GD/SGD or their variants (e.g. Ge et al. (2015); Levy (2016); Fang et al. (2019); Jin et al. (2017; 2018; 2019); Daneshmand et al. (2018);

---

[3]It is known that phase retrieval is nonconvex and has the so-called strict saddle property: (1) every local minimizer $\{w^*, -w^*\}$ is global up to phase, (2) each saddle exhibits negative curvature (see e.g. (Sun et al. (2015; 2016); Chen et al. (2018)))

Staib et al. (2019)). Our work belongs to this category. In this category, perhaps the pioneer works are (Ge et al. (2015)) and (Jin et al. (2017)). Jin et al. (2017) show that explicitly adding isotropic noise in each iteration guarantees that GD escapes saddle points and finds a second order stationary point with high probability. Following (Jin et al. (2017)), Daneshmand et al. (2018) assume that stochastic gradient inherently has a component to escape. Specifically, they make assumption of the Correlated Negative Curvature (CNC) for stochastic gradient $g_t$ so that $\mathbb{E}_t[\langle g_t, v_t \rangle^2] \geq \gamma > 0$. The assumption allows the algorithm to avoid the procedure of perturbing the updates by adding isotropic noise. Our work is motivated by (Daneshmand et al. (2018)) but assumes CNC for the stochastic momentum $m_t$ instead. In Appendix A, we compare the results of our work with the related works.

## 3 MAIN RESULTS

We assume that the gradient $\nabla f$ is $L$-Lipschitz; that is, $f$ is $L$-smooth. Further, we assume that the Hessian $\nabla^2 f$ is $\rho$-Lipschitz. These two properties ensure that $\|\nabla f(w) - \nabla f(w')\| \leq L\|w - w'\|$ and that $\|\nabla^2 f(w) - \nabla^2 f(w')\| \leq \rho\|w - w'\|, \forall w, w'$. The $L$-Lipschitz gradient assumption implies that $|f(w') - f(w) - \langle \nabla f(w), w' - w \rangle| \leq \frac{L}{2}\|w - w'\|^2, \forall w, w'$, while the $\rho$-Lipschitz Hessian assumption implies that $|f(w') - f(w) - \langle \nabla f(w), w' - w \rangle - (w' - w)^\top \nabla^2 f(w)(w' - w)| \leq \frac{\rho}{6}\|w - w'\|^3, \forall w, w'$. Furthermore, we assume that the stochastic gradient has bounded noise $\|\nabla f(w) - \nabla f(w; \xi)\|^2 \leq \sigma^2$ and that the norm of stochastic momentum is bounded so that $\|m_t\| \leq c_m$. We denote $\Pi_i M_i$ as the matrix product of matrices $\{M_i\}$ and we use $\sigma_{max}(M) = \|M\|_2 := \sup_{x \neq 0} \frac{\langle x, Mx \rangle}{\langle x, x \rangle}$ to denote the spectral norm of the matrix $M$.

### 3.1 REQUIRED PROPERTIES WITH EMPIRICAL VALIDATION

Our analysis of stochastic momentum relies on three properties of the stochastic momentum dynamic. These properties are somewhat unusual, but we argue they should hold in natural settings, and later we aim to demonstrate that they hold empirically in a couple of standard problems of interest.

**Definition 2.** *We say that SGD with stochastic momentum satisfies **A**lmost **P**ositively **A**ligned with **G**radient (APAG) [4] if we have*

$$\mathbb{E}_t[\langle \nabla f(w_t), m_t - g_t \rangle] \geq -\frac{1}{2}\|\nabla f(w_t)\|^2. \tag{5}$$

*We say that SGD with stochastic momentum satisfies **A**lmost **P**ositively **C**orrelated with **G**radient (APCG) with parameter $\tau$ if $\exists c' > 0$ such that,*

$$\mathbb{E}_t[\langle \nabla f(w_t), M_t m_t \rangle] \geq -c' \eta \sigma_{max}(M_t)\|\nabla f(w_t)\|^2, \tag{6}$$

*where the PSD matrix $M_t$ is defined as*

$$M_t = (\Pi_{s=1}^{\tau-1} G_{s,t})(\Pi_{s=k}^{\tau-1} G_{s,t}) \quad with \quad G_{s,t} := I - \eta \sum_{j=1}^{s} \beta^{s-j} \nabla^2 f(w_t) = I - \frac{\eta(1-\beta^s)}{1-\beta} \nabla^2 f(w_t)$$

*for any integer $1 \leq k \leq \tau - 1$, and $\eta$ is any step size chosen that guarantees each $G_{s,t}$ is PSD.*

**Definition 3.** *We say that the SGD with momentum exhibits **G**radient **A**lignment or **C**urvature **E**xploitation (GrACE) if $\exists c_h \geq 0$ such that*

$$\mathbb{E}_t[\eta \langle \nabla f(w_t), g_t - m_t \rangle + \frac{\eta^2}{2} m_t^\top \nabla^2 f(w_t) m_t] \leq \eta^2 c_h. \tag{7}$$

APAG requires that the momentum term $m_t$ must, in expectation, not be significantly misaligned with the gradient $\nabla f(w_t)$. This is a very natural condition when one sees that the momentum term is acting as a biased estimate of the gradient of the deterministic $f$. APAG demands that the bias can not be too large relative to the size of $\nabla f(w_t)$. Indeed this property is only needed in our analysis when the gradient is large (i.e. $\|\nabla f(w_t)\| \geq \epsilon$) as it guarantees that the algorithm makes progress; our analysis does not require APAG holds when gradient is small.

APCG is a related property, but requires that the current momentum term $m_t$ is almost positively correlated with the the gradient $\nabla f(w_t)$, but *measured in the Mahalanobis norm induced by $M_t$.* It

---

[4]Note that our analysis still go through if one replaces $\frac{1}{2}$ on r.h.s. of (5) with any larger number $c < 1$; the resulted iteration complexity would be only a constant multiple worse.

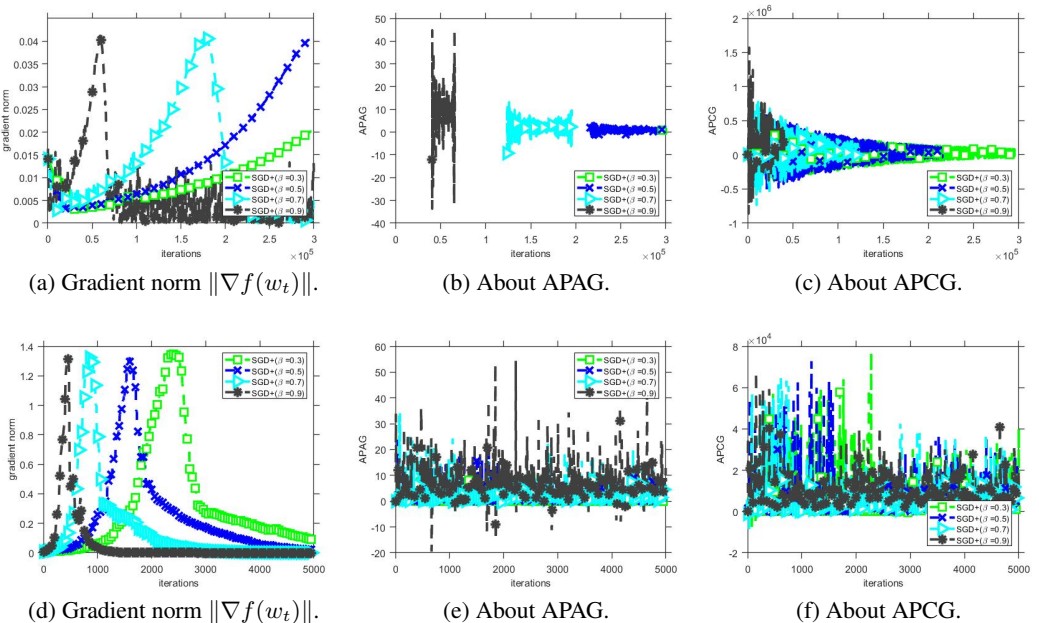

Figure 3: Plots of the related properties. Sub-figures on the top row are regarding solving (3) and sub-figures on the bottom row are regarding solving (4) (phase retrieval). Note that the function value/relative distance to $w^*$ are plotted on Figure 2. Above, sub-figures (a) and (d): We plot the gradient norms versus iterations. Sub-figures (b) and (e): We plot the values of $\langle \nabla f(w_t), m_t - g_t \rangle / \|\nabla f(w_t)\|^2$ versus iterations. For (b), we only report them when the gradient is large ($\|\nabla f(w_t)\| \geq 0.02$). It shows that the value is large than $-0.5$ except the transition. For (e), we observe that the value is almost always nonnegative. Sub-figures (c) and (f): We plot the value of $\langle \nabla f(w_t), M_t m_t \rangle / \eta \sigma_{max}(M_t) \|\nabla f(w_t)\|^2$. For (c), we let $M_t = (\Pi_{s=1}^{3 \times 10^5} G_{s,t})(\Pi_{s=1}^{3 \times 10^5} G_{s,t})$ and we only report the values when the update is in the region of saddle points. For (f), we let $M_t = (\Pi_{s=1}^{500} G_{s,t})(\Pi_{s=1}^{500} G_{s,t})$ and we observe that the value is almost always nonnegative. The figures implies that SGD with momentum has APAG and APCG properties in the experiments. Furthermore, an interesting observation is that, for the phase retrieval problem, the expected values might actually be nonnegative.

may appear to be an unusual object, but one can view the PSD matrix $M_t$ as measuring something about the local curvature of the function with respect to the trajectory of the SGD with momentum dynamic. We will show that this property holds empirically on two natural problems for a reasonable constant $c'$. APCG is only needed in our analysis when the update is in a saddle region with significant negative curvature, $\|\nabla f(w)\| \leq \epsilon$ and $\lambda_{\min}(\nabla^2 f(w)) \leq -\epsilon$. Our analysis does not require APCG holds when the gradient is large or the update is at an $(\epsilon, \epsilon)$-second order stationary point.

For GrACE, the first term on l.h.s of (7) measures the alignment between stochastic momentum $m_t$ and the gradient $\nabla f(w_t)$, while the second term on l.h.s measures the curvature exploitation. The first term is small (or even negative) when the stochastic momentum $m_t$ is aligned with the gradient $\nabla f(w_t)$, while the second term is small (or even negative) when the stochastic momentum $m_t$ can exploit a negative curvature (i.e. the subspace of eigenvectors that corresponds to the negative eigenvalues of the Hessian $\nabla^2 f(w_t)$ if exists). Overall, a small sum of the two terms (and, consequently, a small $c_h$) allows one to bound the function value of the next iterate (see Lemma 8).

On Figure 3, we report some quantities related to APAG and APCG as well as the gradient norm when solving the previously discussed problems (3) and (4) using SGD with momentum. We also report a quantity regarding GrACE on Figure 4 in the appendix.

## 3.2 CONVERGENCE RESULTS

The high level idea of our analysis follows as a similar template to (Jin et al. (2017); Daneshmand et al. (2018); Staib et al. (2019)). Our proof is structured into three cases: either (a) $\|\nabla f(w)\| \geq \epsilon$, or (b) $\|\nabla f(w)\| \leq \epsilon$ and $\lambda_{\min}(\nabla^2 f(w)) \leq -\epsilon$, or otherwise (c) $\|\nabla f(w)\| \leq \epsilon$ and $\lambda_{\min}(\nabla^2 f(w)) \geq -\epsilon$,

---

**Algorithm 2:** SGD with stochastic heavy ball momentum

1: Required: Step size parameters $r$ and $\eta$, momentum parameter $\beta$, and period parameter $\mathcal{T}_{thred}$.
2: Init: $w_0 \in \mathbb{R}^d$ and $m_{-1} = 0 \in \mathbb{R}^d$.
3: **for** $t = 0$ to $T$ **do**
4:     Get stochastic gradient $g_t$ at $w_t$, and set stochastic momentum $m_t := \beta m_{t-1} + g_t$.
5:     Set learning rate: $\hat{\eta} := \eta$ **unless** $(t \bmod \mathcal{T}_{thred}) = 0$ in which case $\hat{\eta} := r$
6:     $w_{t+1} = w_t - \hat{\eta} m_t$.
7: **end for**

---

meaning we have arrived in a second-order stationary region. The precise algorithm we analyze is Algorithm 2, which identical to Algorithm 1 except that we boost the step size to a larger value $r$ on occasion. We will show that the algorithm makes progress in cases (a) and (b). In case (c), when the goal has already been met, further execution of the algorithm only weakly hurts progress. Ultimately, we prove that a second order stationary point is arrived at with high probability. While our proof borrows tools from (Daneshmand et al. (2018); Staib et al. (2019)), much of the momentum analysis is entirely novel to our knowledge.

**Theorem 1.** *Assume that the stochastic momentum satisfies CNC. Set [5] $r = O(\epsilon^2)$, $\eta = O(\epsilon^5)$, and $\mathcal{T}_{thred} = \frac{c(1-\beta)}{\eta\epsilon} \log(\frac{Lc_m\sigma^2\rho c'c_h}{(1-\beta)\delta\gamma\epsilon}) = O((1-\beta)\log(\frac{Lc_m\sigma^2\rho c'c_h}{(1-\beta)\delta\gamma\epsilon})\epsilon^{-6})$ for some constant $c > 0$. If SGD with momentum (Algorithm 2) has APAG property when gradient is large ($\|\nabla f(w)\| \geq \epsilon$), $APCG_{\mathcal{T}_{thred}}$ property when it enters a region of saddle points that exhibits a negative curvature ($\|\nabla f(w)\| \leq \epsilon$ and $\lambda_{\min}(\nabla^2 f(w)) \leq -\epsilon$), and GrACE property throughout the iterations, then it reaches an $(\epsilon, \epsilon)$ second order stationary point in $T = 2\mathcal{T}_{thred}(f(w_0) - \min_w f(w))/(\delta\mathcal{F}_{thred}) = O((1-\beta)\log(\frac{Lc_m\sigma^2\rho c'c_h}{(1-\beta)\delta\gamma\epsilon})\epsilon^{-10})$ iterations with high probability $1 - \delta$, where $\mathcal{F}_{thred} = O(\epsilon^4)$.*

The theorem implies the advantage of using stochastic momentum for SGD. Higher $\beta$ leads to reaching a second order stationary point faster. As we will show in the following, this is due to that higher $\beta$ enables escaping the saddle points faster. In Subsection 3.2.1, we provide some key details of the proof of Theorem 1. The interested reader can read a high-level sketch of the proof, as well as the detailed version, in Appendix G.

**Remark 1: (constraints on $\beta$)** We also need some minor constraints on $\beta$ so that $\beta$ cannot be too close to 1. They are 1) $L(1-\beta)^3 > 1$, 2) $\sigma^2(1-\beta)^3 > 1$, 3) $c'(1-\beta)^2 > 1$, 4) $\eta \leq \frac{1-\beta}{L}$, 5) $\eta \leq \frac{1-\beta}{\epsilon}$, and 6) $\mathcal{T}_{thred} \geq 1 + \frac{2\beta}{1-\beta}$. Please see Appendix E.1 for the details and discussions.

**Remark 2: (escaping saddle points)** Note that Algorithm 2 reduces to CNC-SGD of Daneshmand et al. (2018) when $\beta = 0$ (i.e. without momentum). Therefore, let us compare the results. We show that the escape time of Algorithm 2 is $T_{thred} := \tilde{O}\left(\frac{(1-\beta)}{\eta\epsilon}\right)$ (see Appendix E.3.3, especially (81-82)). On the other hand, for CNC-SGD, based on Table 3 in their paper, is $T_{thred} = \tilde{O}\left(\frac{1}{\eta\epsilon}\right)$. One can clearly see that $T_{thred}$ of our result has a dependency $1 - \beta$, which makes it smaller than that of Daneshmand et al. (2018) for any same $\eta$ and consequently demonstrates escaping saddle point faster with momentum.

**Remark 3: (finding a second order stationary point)** Denote $\ell$ a number such that $\forall t, \|g_t\| \leq \ell$. In Appendix G.3, we show that in the high momentum regime where $(1 - \beta) << \frac{\rho^2\ell^{10}}{c_m^9 c_h^2 c'}$, Algorithm 2 is strictly better than CNC-SGD of Daneshmand et al. (2018), which means that a higher momentum can help find a second order stationary point faster. Empirically, we find out that $c' \approx 0$ (Figure 3) and $c_h \approx 0$ (Figure 4) in the phase retrieval problem, so the condition is easily satisfied for a wide range of $\beta$.

### 3.2.1 ESCAPING SADDLE POINTS

In this subsection, we analyze the process of escaping saddle points by SGD with momentum. Denote $t_0$ any time such that $(t_0 \bmod \mathcal{T}_{thred}) = 0$. Suppose that it enters the region exhibiting a small

---

[5]See Table 3 in Appendix E for the precise expressions of the parameters. Here, we hide the parameters' dependencies on $\gamma$, $L$, $c_m$, $c'$, $\sigma^2$, $\rho$, $c_h$, and $\delta$. W.l.o.g, we also assume that $c_m$, $L$, $\sigma^2$, $c'$, $c_h$, and $\rho$ are not less than one and $\epsilon \leq 1$.

gradient but a large negative eigenvalue of the Hessian (i.e. $\|\nabla f(w_{t_0})\| \leq \epsilon$ and $\lambda_{\min}(\nabla^2 f(w_{t_0})) \leq -\epsilon$). We want to show that it takes at most $\mathcal{T}_{thred}$ iterations to escape the region and whenever it escapes, the function value decreases at least by $\mathcal{F}_{thred} = O(\epsilon^4)$ on expectation, where the precise expression of $\mathcal{F}_{thred}$ will be determined later in Appendix E. The technique that we use is proving by contradiction. Assume that the function value on expectation does not decrease at least $\mathcal{F}_{thred}$ in $\mathcal{T}_{thred}$ iterations. Then, we get an upper bound of the expected distance $\mathbb{E}_{t_0}[\|w_{t_0+\mathcal{T}_{thred}} - w_{t_0}\|^2] \leq C_{\text{upper}}$. Yet, by leveraging the negative curvature, we also show a lower bound of the form $\mathbb{E}_{t_0}[\|w_{t_0+\mathcal{T}_{thred}} - w_{t_0}\|^2] \geq C_{\text{lower}}$. The analysis will show that the lower bound is larger than the upper bound (namely, $C_{\text{lower}} > C_{\text{upper}}$), which leads to the contradiction and concludes that the function value must decrease at least $\mathcal{F}_{thred}$ in $\mathcal{T}_{thred}$ iterations on expectation. Since $\mathcal{T}_{thred} = O((1-\beta)\log(\frac{1}{(1-\beta)\epsilon})\epsilon^6)$, the dependency on $\beta$ suggests that larger $\beta$ can leads to smaller $\mathcal{T}_{thred}$, which implies that larger momentum helps in escaping saddle points faster.

Lemma 1 below provides an upper bound of the expected distance. The proof is in Appendix C.

**Lemma 1.** *Denote $t_0$ any time such that $(t_0 \mod \mathcal{T}_{thred}) = 0$. Suppose that $\mathbb{E}_{t_0}[f(w_{t_0}) - f(w_{t_0+t})] \leq \mathcal{F}_{thred}$ for any $0 \leq t \leq \mathcal{T}_{thred}$. Then, $\mathbb{E}_{t_0}[\|w_{t_0+t} - w_{t_0}\|^2] \leq C_{\text{upper},t} := \frac{8\eta t\left(\mathcal{F}_{thred}+2r^2 c_h + \frac{\rho}{3}r^3 c_m^3\right)}{(1-\beta)^2} + 8\eta^2 \frac{t\sigma^2}{(1-\beta)^2} + 4\eta^2\left(\frac{\beta}{1-\beta}\right)^2 c_m^2 + 2r^2 c_m^2.$*

We see that $C_{\text{upper,t}}$ in Lemma 1 is monotone increasing with $t$, so we can define $C_{\text{upper}} := C_{\text{upper},\mathcal{T}_{thred}}$. Now let us switch to obtaining the lower bound of $\mathbb{E}_{t_0}[\|w_{t_0+\mathcal{T}_{thred}} - w_{t_0}\|^2]$. The key to get the lower bound comes from the recursive dynamics of SGD with momentum.

**Lemma 2.** *Denote $t_0$ any time such that $(t_0 \mod \mathcal{T}_{thred}) = 0$. Let us define a quadratic approximation at $w_{t_0}$, $Q(w) := f(w_{t_0}) + \langle w - w_{t_0}, \nabla f(w_{t_0})\rangle + \frac{1}{2}(w - w_{t_0})^\top H(w - w_{t_0})$, where $H := \nabla^2 f(w_{t_0})$. Also, define $G_s := (I - \eta\sum_{k=1}^{s}\beta^{s-k}H)$. Then we can write $w_{t_0+t} - w_{t_0}$ exactly using the following decomposition.*

$$\overbrace{\left(\Pi_{j=1}^{t-1}G_j\right)\left(-rm_{t_0}\right)}^{q_{v,t-1}} + \overbrace{\eta(-1)\sum_{s=1}^{t-1}\left(\Pi_{j=s+1}^{t-1}G_j\right)\beta^s m_{t_0}}^{q_{m,t-1}}$$

$$+ \quad \overbrace{\eta(-1)\sum_{s=1}^{t-1}\left(\Pi_{j=s+1}^{t-1}G_j\right)\sum_{k=1}^{s}\beta^{s-k}\left(\nabla f(w_{t_0+k}) - \nabla Q(w_{t_0+s})\right)}^{q_{q,t-1}}$$

$$+ \quad \overbrace{\eta(-1)\sum_{s=1}^{t-1}\left(\Pi_{j=s+1}^{t-1}G_j\right)\sum_{k=1}^{s}\beta^{s-k}\nabla f(w_{t_0})}^{q_{w,t-1}} + \overbrace{\eta(-1)\sum_{s=1}^{t-1}\left(\Pi_{j=s+1}^{t-1}G_j\right)\sum_{k=1}^{s}\beta^{s-k}\xi_{t_0+k}}^{q_{\xi,t-1}}.$$

The proof of Lemma 2 is in Appendix D. Furthermore, we will use the quantities $q_{v,t-1}, q_{m,t-1}, q_{q,t-1}, q_{w,t-1}, q_{\xi,t-1}$ as defined above throughout the analysis.

**Lemma 3.** *Following the notations of Lemma 2, we have that*

$$\mathbb{E}_{t_0}[\|w_{t_0+t} - w_{t_0}\|^2] \geq \mathbb{E}_{t_0}[\|q_{v,t-1}\|^2] + 2\eta\mathbb{E}_{t_0}[\langle q_{v,t-1}, q_{m,t-1} + q_{q,t-1} + q_{w,t-1} + q_{\xi,t-1}\rangle] =: C_{lower}.$$

We are going to show that the dominant term in the lower bound of $\mathbb{E}_{t_0}[\|w_{t_0+t} - w_{t_0}\|^2]$ is $\mathbb{E}_{t_0}[\|q_{v,t-1}\|^2]$, which is the critical component for ensuring that the lower bound is larger than the upper bound of the expected distance.

**Lemma 4.** *Denote $\theta_j := \sum_{k=1}^{j}\beta^{j-k} = \sum_{k=1}^{j}\beta^{k-1}$ and $\lambda := -\lambda_{\min}(H)$. Following the conditions and notations in Lemma 1 and Lemma 2, we have that*

$$\mathbb{E}_{t_0}[\|q_{v,t-1}\|^2] \geq \left(\Pi_{j=1}^{t-1}(1 + \eta\theta_j\lambda)\right)^2 r^2\gamma. \tag{8}$$

*Proof.* We know that $\lambda_{\min}(H) \leq -\epsilon < 0$. Let $v$ be the eigenvector of the Hessian $H$ with unit norm that corresponds to $\lambda_{\min}(H)$ so that $Hv = \lambda_{\min}(H)v$. We have $(I - \eta H)v = v - \eta\lambda_{\min}(H)v =$

$(1 - \eta\lambda_{\min}(H))v$. Then,

$$
\begin{aligned}
\mathbb{E}_{t_0}[\|q_{v,t-1}\|^2] &\stackrel{(a)}{=} \mathbb{E}_{t_0}[\|q_{v,t-1}\|^2\|v\|^2] \stackrel{(b)}{\geq} \mathbb{E}_{t_0}[\langle q_{v,t-1}, v\rangle^2] \stackrel{(c)}{=} \mathbb{E}_{t_0}[\langle(\Pi_{j=1}^{t-1}G_j)rm_{t_0}, v\rangle^2] \\
&\stackrel{(d)}{=} \mathbb{E}_{t_0}[\langle(\Pi_{j=1}^{t-1}(I - \eta\theta_j H))rm_{t_0}, v\rangle^2] = \mathbb{E}_{t_0}\langle(\Pi_{j=1}^{t-1}(1 - \eta\theta_j\lambda_{\min}(H)))rm_{t_0}, v\rangle^2] \\
&\stackrel{(e)}{\geq} \left(\Pi_{j=1}^{t-1}(1 + \eta\theta_j\lambda)\right)^2 r^2\gamma,
\end{aligned}
\tag{9}
$$

where $(a)$ is because $v$ is with unit norm, $(b)$ is by Cauchy–Schwarz inequality, $(c)$, $(d)$ are by the definitions, and $(e)$ is by the CNC assumption so that $\mathbb{E}_{t_0}[\langle m_{t_0}, v\rangle^2] \geq \gamma$. □

Observe that the lower bound in (8) is monotone increasing with $t$ and the momentum parameter $\beta$. Moreover, it actually grows exponentially in $t$. To get the contradiction, we have to show that the lower bound is larger than the upper bound. By Lemma 1 and Lemma 3, it suffices to prove the following lemma. We provide its proof in Appendix E.

**Lemma 5.** *Let $\mathcal{F}_{thred} = O(\epsilon^4)$ and $\eta^2\mathcal{T}_{thred} \leq r^2$. By following the conditions and notations in Theorem 1, Lemma 1 and Lemma 2, we conclude that if SGD with momentum (Algorithm 2) has the APCG property, then we have that $C_{lower} := \mathbb{E}_{t_0}[\|q_{v,\mathcal{T}_{thred}-1}\|^2] + 2\eta\mathbb{E}_{t_0}[\langle q_{v,\mathcal{T}_{thred}-1}, q_{m,\mathcal{T}_{thred}-1} + q_{q,\mathcal{T}_{thred}-1} + q_{w,\mathcal{T}_{thred}-1} + q_{\xi,\mathcal{T}_{thred}-1}\rangle] > C_{upper}$.*

## 4 CONCLUSION

In this paper, we identify three properties that guarantee SGD with momentum in reaching a second-order stationary point faster by a higher momentum, which justifies the practice of using a large value of momentum parameter $\beta$. We show that a greater momentum leads to escaping strict saddle points faster due to that SGD with momentum recursively enlarges the projection to an escape direction. However, how to make sure that SGD with momentum has the three properties is not very clear. It would be interesting to identify conditions that guarantee SGD with momentum to have the properties. Perhaps a good starting point is understanding why the properties hold in phase retrieval. We believe that our results shed light on understanding the recent success of SGD with momentum in non-convex optimization and deep learning.

## ACKNOWLEDGMENTS

We gratefully acknowledge financial support from NSF IIS awards 1910077 and 1453304.

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

## A    LITERATURE SURVEY

**Heavy ball method:** The heavy ball method was originally proposed by Polyak (1964). It has been observed that this algorithm, even in the deterministic setting, provides no convergence speedup over standard gradient descent, except in some highly structure cases such as convex quadratic objectives where an "accelerated" rate is possible (Lessard et al. (2016); Goh (2017)). In recent years, some works make some efforts in analyzing heavy ball method for other classes of optimization problems besides the quadratic functions. For example, Ghadimi et al. (2015) prove an $O(1/T)$ ergodic convergence rate when the problem is smooth convex, while Sun et al. (2019) provide a non-ergodic convergence rate for certain classes of convex problems. Ochs et al. (2014) combine the technique of forward-backward splitting with heavy ball method for a specific class of nonconvex optimization problem. For stochastic heavy ball method, Loizou & Richtárik (2017) analyze a class of linear regression problems and shows a linear convergence rate of stochastic momentum, in which the linear regression problems actually belongs to the case of strongly convex quadratic functions. Other works includes (Gadat et al. (2016)), which shows almost sure convergence to the critical points by stochastic heavy ball for general non-convex coercive functions. Yet, the result does not show any advantage of stochastic heavy ball over other optimization algorithms like SGD. Can et al. (2019) show an accelerated linear convergence to a stationary distribution under Wasserstein distance for strongly convex quadratic functions by SGD with stochastic heavy ball momentum. Yang et al. (2018) provide a unified analysis of stochastic heavy ball momentum and Nesterov's momentum for smooth non-convex objective functions. They show that the expected gradient norm converges at rate $O(1/\sqrt{t})$. Yet, the rate is not better than that of the standard SGD. We are also aware of the works (Ghadimi & Lan (2016; 2013)), which propose some variants of stochastic accelerated algorithms with first order stationary point guarantees. Yet, the framework in (Ghadimi & Lan (2016; 2013)) does not capture the stochastic heavy ball momentum used in practice. There is also a negative result about the heavy ball momentum. Kidambi et al. (2018) show that for a specific strongly convex and

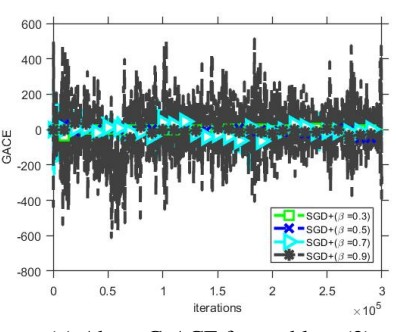 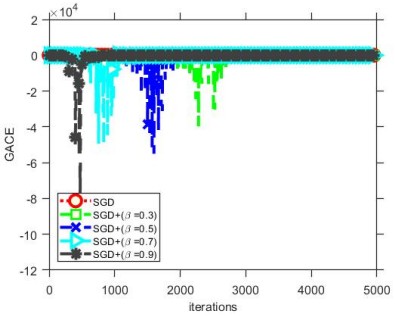

(a) About GrACE for problem (3).

(b) About GrACE for problem (4) (phase retrieval).

Figure 4: Plot regarding the GrACE property. We plot the values of $\frac{\eta \langle \nabla f(w_t), g_t - m_t \rangle + \frac{1}{2}\eta^2 m_t^\top H_t m_t}{\eta^2}$ versus iterations. An interesting observation is that the value is well upper-bounded by zero for the phase retrieval problem. The results imply that the constant $c_h$ is indeed small.

strongly smooth problem, SGD with heavy ball momentum fails to achieving the best convergence rate while some algorithms can.

**Reaching a second order stationary point:** As we mentioned earlier, there are many works aim at reaching a second order stationary point. We classify them into two categories: specialized algorithms and simple GD/SGD variants. Specialized algorithms are those designed to exploit the negative curvature explicitly and escape saddle points faster than the ones without the explicit exploitation (e.g. Carmon et al. (2018); Agarwal et al. (2017); Allen-Zhu & Li (2018); Xu et al. (2018)). Simple GD/SGD variants are those with minimal tweaks of standard GD/SGD or their variants (e.g. Ge et al. (2015); Levy (2016); Fang et al. (2019); Jin et al. (2017; 2018; 2019); Daneshmand et al. (2018); Staib et al. (2019)). Our work belongs to this category. In this category, perhaps the pioneer works are (Ge et al. (2015)) and (Jin et al. (2017)). Jin et al. (2017) show that explicitly adding isotropic noise in each iteration guarantees that GD escapes saddle points and finds a second order stationary point with high probability. Following (Jin et al. (2017)), Daneshmand et al. (2018) assume that stochastic gradient inherently has a component to escape. Specifically, they make assumption of the Correlated Negative Curvature (CNC) for stochastic gradient $g_t$ so that $\mathbb{E}_t[\langle g_t, v_t \rangle^2] \geq \gamma > 0$. The assumption allows the algorithm to avoid the procedure of perturbing the updates by adding isotropic noise. Our work is motivated by (Daneshmand et al. (2018)) but assumes CNC for the stochastic momentum $m_t$ instead. Very recently, Jin et al. (2019) consider perturbing the update of SGD and provide a second order guarantee. Staib et al. (2019) consider a variant of RMSProp (Tieleman & Hinton (2012)), in which the gradient $g_t$ is multiplied by a preconditioning matrix $G_t$ and the update is $w_{t+1} = w_t - G_t^{-1/2} g_t$. The work shows that the algorithm can help in escaping saddle points faster compared to the standard SGD under certain conditions. Fang et al. (2019) propose average-SGD, in which a suffix averaging scheme is conducted for the updates. They also assume an inherent property of stochastic gradients that allows SGD to escape saddle points.

We summarize the iteration complexity results of the related works for simple SGD variants on Table 1. [6] The readers can see that the iteration complexity of (Fang et al. (2019)) and (Jin et al. (2019)) are better than (Daneshmand et al. (2018); Staib et al. (2019)) and our result. So, we want to explain the results and clarify the differences. First, we focus on explaining why the popular algorithm, SGD with heavy ball momentum, works well in practice, which is without the suffix averaging scheme used in (Fang et al. (2019)) and is without the explicit perturbation used in (Jin et al. (2019)). Specifically, we focus on studying the effect of stochastic heavy ball momentum and showing the advantage of using it. Furthermore, our analysis framework is built on the work of (Daneshmand et al. (2018)). We believe that, based on the insight in our work, one can also show the advantage of stochastic momentum by modifying the assumptions and algorithms in (Fang et al. (2019)) or (Jin et al. (2019)) and consequently get a better dependency on $\epsilon$.

---

[6]We follow the work (Daneshmand et al. (2018)) for reaching an $(\epsilon, \epsilon)$-stationary point, while some works are for an $(\epsilon, \sqrt{\epsilon})$-stationary point. We translate them into the complexity of getting an $(\epsilon, \epsilon)$-stationary point.

| Algorithm | Complexity |
|---|---|
| Perturbed SGD (Ge et al. (2015)) | $\mathcal{O}(\epsilon^{-16})$ |
| Average-SGD (Fang et al. (2019)) | $\mathcal{O}(\epsilon^{-7})$ |
| Perturbed SGD (Jin et al. (2019)) | $\mathcal{O}(\epsilon^{-8})$ |
| CNC-SGD (Daneshmand et al. (2018)) | $\mathcal{O}(\epsilon^{-10})$ |
| Adaptive SGD (Staib et al. (2019)) | $\mathcal{O}(\epsilon^{-10})$ |
| SGD+momentum (this work) | $\mathcal{O}((1-\beta)\log(\frac{1}{(1-\beta)\epsilon})\epsilon^{-10})$ |

Table 1: Iteration complexity to find an $(\epsilon, \epsilon)$ second-order stationary point .

# B  Lemma 6, 7, and 8

In the following, Lemma 7 says that under the APAG property, when the gradient norm is large, on expectation SGD with momentum decreases the function value by a constant and consequently makes progress. On the other hand, Lemma 8 upper-bounds the increase of function value of the next iterate (if happens) by leveraging the GrACE property.

**Lemma 6.** *If SGD with momentum has the APAG property, then, considering the update step* $w_{t+1} = w_t - \eta m_t$, *we have that* $\mathbb{E}_t[f(w_{t+1})] \leq f(w_t) - \frac{\eta}{2}\|\nabla f(w_t)\|^2 + \frac{L\eta^2 c_m^2}{2}$.

*Proof.* By the $L$-smoothness assumption,

$$f(w_{t+1}) \leq f(w_t) - \eta\langle \nabla f(w_t), m_t\rangle + \frac{L\eta^2}{2}\|m_t\|^2$$

$$\leq f(w_t) - \eta\langle \nabla f(w_t), g_t\rangle - \eta\langle \nabla f(w_t), m_t - g_t\rangle + \frac{L\eta^2 c_m^2}{2}. \tag{10}$$

Taking the expectation on both sides. We have

$$\mathbb{E}_t[f(w_{t+1})] \leq f(w_t) - \eta\|\nabla f(w_t)\|^2 - \eta\mathbb{E}_t[\langle \nabla f(w_t), m_t - g_t\rangle] + \frac{L\eta^2 c_m^2}{2}$$

$$\leq f(w_t) - \frac{\eta}{2}\|\nabla f(w_t)\|^2 + \frac{L\eta^2 c_m^2}{2}. \tag{11}$$

where we use the APAG property in the last inequality.

$\square$

**Lemma 7.** *Assume that the step size* $\eta$ *satisfies* $\eta \leq \frac{\epsilon^2}{8Lc_m^2}$. *If SGD with momentum has the APAG property, then, considering the update step* $w_{t+1} = w_t - \eta m_t$, *we have that* $\mathbb{E}_t[f(w_{t+1})] \leq f(w_t) - \frac{\eta}{4}\epsilon^2$ *when* $\|\nabla f(w_t)\| \geq \epsilon$.

*Proof.* $\mathbb{E}_t[f(w_{t+1}) - f(w_t)] \overset{Lemma\ 6}{\leq} -\frac{\eta}{2}\|\nabla f(w_t)\|^2 + \frac{L\eta^2 c_m^2}{2} \overset{\|\nabla f(w_t)\|\geq\epsilon}{\leq} -\frac{\eta}{2}\epsilon^2 + \frac{L\eta^2 c_m^2}{2} \leq -\frac{\eta}{4}\epsilon^2$, where the last inequality is due to the constraint of $\eta$. $\square$

**Lemma 8.** *If SGD with momentum has the GrACE property, then, considering the update step* $w_{t+1} = w_t - \eta m_t$, *we have that* $\mathbb{E}_t[f(w_{t+1})] \leq f(w_t) + \eta^2 c_h + \frac{\rho\eta^3}{6}c_m^3$.

*Proof.* Consider the update rule $w_{t+1} = w_t - \eta m_t$, where $m_t$ represents the stochastic momentum and $\eta$ is the step size. By $\rho$-Lipschitzness of Hessian, we have $f(w_{t+1}) \leq f(w_t) - \eta\langle \nabla f(w_t), g_t\rangle + \eta\langle \nabla f(w_t), g_t - m_t\rangle + \frac{\eta^2}{2}m_t^\top \nabla^2 f(w_t)m_t + \frac{\rho\eta^3}{6}\|m_t\|^3$. Taking the conditional expectation, one has

$$\mathbb{E}_t[f(w_{t+1})] \leq f(w_t) - \mathbb{E}_t[\eta\|\nabla f(w_t)\|^2] + \mathbb{E}_t[\eta\langle \nabla f(w_t), g_t - m_t\rangle + \frac{\eta^2}{2}m_t^\top \nabla^2 f(w_t)m_t + \frac{\rho\eta^3}{6}c_m^3.$$

$$\leq f(w_t) + 0 + \eta^2 c_h + \frac{\rho\eta^3}{6}c_m^3.$$

$$\tag{12}$$

$\square$

## C    PROOF OF LEMMA 1

**Lemma 1** *Denote $t_0$ any time such that $(t_0 \mod \mathcal{T}_{thred}) = 0$. Suppose that $\mathbb{E}_{t_0}[f(w_{t_0}) - f(w_{t_0+t})] \leq \mathcal{F}_{thred}$ for any $0 \leq t \leq \mathcal{T}_{thred}$. Then,*

$$\mathbb{E}_{t_0}[\|w_{t_0+t} - w_{t_0}\|^2] \leq C_{upper,t}$$
$$:= \frac{8\eta t\big(\mathcal{F}_{thred} + 2r^2 c_h + \frac{\rho}{3}r^3 c_m^3\big)}{(1-\beta)^2} + 8\eta^2 \frac{t\sigma^2}{(1-\beta)^2} + 4\eta^2 \big(\frac{\beta}{1-\beta}\big)^2 c_m^2 + 2r^2 c_m^2. \tag{13}$$

*Proof.* Recall that the update is $w_{t_0+1} = w_{t_0} - rm_{t_0}$, and $w_{t_0+t} = w_{t_0+t-1} - \eta m_{t_0+t-1}$, for $t > 1$. We have that

$$\|w_{t_0+t} - w_{t_0}\|^2 \leq 2(\|w_{t_0+t} - w_{t_0+1}\|^2 + \|w_{t_0+1} - w_{t_0}\|^2) \leq 2\|w_{t_0+t} - w_{t_0+1}\|^2 + 2r^2 c_m^2, \tag{14}$$

where the first inequality is by the triangle inequality and the second one is due to the assumption that $\|m_t\| \leq c_m$ for any $t$. Now let us denote

- $\alpha_s := \sum_{j=0}^{t-1-s} \beta^j$

- $A_{t-1} := \sum_{s=1}^{t-1} \alpha_s$

and let us rewrite $g_t = \nabla f(w_t) + \xi_t$, where $\xi_t$ is the zero-mean noise. We have that

$$\mathbb{E}_{t_0}[\|w_{t_0+t} - w_{t_0+1}\|^2] = \mathbb{E}_{t_0}[\|\sum_{s=1}^{t-1} -\eta m_{t_0+s}\|^2] = \mathbb{E}_{t_0}[\eta^2 \|\sum_{s=1}^{t-1} \big((\sum_{j=1}^{s} \beta^{s-j} g_{t_0+j}) + \beta^s m_{t_0}\big)\|^2]$$

$$\leq \mathbb{E}_{t_0}[2\eta^2 \|\sum_{s=1}^{t-1}\sum_{j=1}^{s} \beta^{s-j} g_{t_0+j}\|^2 + 2\eta^2 \|\sum_{s=1}^{t-1} \beta^s m_{t_0}\|^2]$$

$$\leq \mathbb{E}_{t_0}[2\eta^2 \|\sum_{s=1}^{t-1}\sum_{j=1}^{s} \beta^{s-j} g_{t_0+j}\|^2] + 2\eta^2 \big(\frac{\beta}{1-\beta}\big)^2 c_m^2$$

$$= \mathbb{E}_{t_0}[2\eta^2 \|\sum_{s=1}^{t-1} \alpha_s g_{t_0+s}\|^2] + 2\eta^2 \big(\frac{\beta}{1-\beta}\big)^2 c_m^2$$

$$= \mathbb{E}_{t_0}[2\eta^2 \|\sum_{s=1}^{t-1} \alpha_s \big(\nabla f(w_{t_0+s}) + \xi_{t_0+s}\big)\|^2] + 2\eta^2 \big(\frac{\beta}{1-\beta}\big)^2 c_m^2$$

$$\leq \mathbb{E}_{t_0}[4\eta^2 \|\sum_{s=1}^{t-1} \alpha_s \nabla f(w_{t_0+s})\|^2] + \mathbb{E}_{t_0}[4\eta^2 \|\sum_{s=1}^{t-1} \alpha_s \xi_{t_0+s}\|^2] + 2\eta^2 \big(\frac{\beta}{1-\beta}\big)^2 c_m^2. \tag{15}$$

To proceed, we need to upper bound $\mathbb{E}_{t_0}[4\eta^2 \|\sum_{s=1}^{t-1} \alpha_s \nabla f(w_{t_0+s})\|^2]$. We have that

$$\mathbb{E}_{t_0}[4\eta^2 \|\sum_{s=1}^{t-1} \alpha_s \nabla f(w_{t_0+s})\|^2] \overset{(a)}{\leq} \mathbb{E}_{t_0}[4\eta^2 A_{t-1}^2 \sum_{s=1}^{t-1} \frac{\alpha_s}{A_{t-1}} \|\nabla f(w_{t_0+s})\|^2]$$

$$\overset{(b)}{\leq} \mathbb{E}_{t_0}[4\eta^2 \frac{A_{t-1}}{1-\beta} \sum_{s=1}^{t-1} \|\nabla f(w_{t_0+s})\|^2] \overset{(c)}{\leq} \mathbb{E}_{t_0}[4\eta^2 \frac{t}{(1-\beta)^2} \sum_{s=1}^{t-1} \|\nabla f(w_{t_0+s})\|^2]. \tag{16}$$

where $(a)$ is by Jensen's inequality, $(b)$ is by $\max_s \alpha_s \leq \frac{1}{1-\beta}$, and $(c)$ is by $A_{t-1} \leq \frac{t}{1-\beta}$. Now let us switch to bound the other term.

$$
\begin{aligned}
\mathbb{E}_{t_0}[4\eta^2\|\sum_{s=1}^{t-1}\alpha_s\xi_{t_0+s}\|^2] &= 4\eta^2\big(\mathbb{E}_{t_0}[\sum_{i\neq j}\alpha_i\alpha_j\xi_{t_0+i}^\top\xi_{t_0+j}] + \mathbb{E}_{t_0}[\sum_{s=1}^{t-1}\alpha_s^2\xi_{t_0+s}^\top\xi_{t_0+s}]\big) \\
&\overset{(a)}{=} 4\eta^2\big(0 + \mathbb{E}_{t_0}[\sum_{s=1}^{t-1}\alpha_s^2\xi_{t_0+s}^\top\xi_{t_0+s}]\big), \\
&\overset{(b)}{\leq} 4\eta^2\frac{t\sigma^2}{(1-\beta)^2}.
\end{aligned}
\tag{17}
$$

where $(a)$ is because $\mathbb{E}_{t_0}[\xi_{t_0+i}^\top\xi_{t_0+j}] = 0$ for $i \neq j$, $(b)$ is by that $\|\xi_t\|^2 \leq \sigma^2$ and $\max_t \alpha_t \leq \frac{1}{1-\beta}$. Combining (14), (15), (16), (17),

$$
\mathbb{E}_{t_0}[\|w_{t_0+t} - w_{t_0}\|^2] \leq \mathbb{E}_{t_0}[8\eta^2\frac{t}{(1-\beta)^2}\sum_{s=1}^{t-1}\|\nabla f(w_{t_0+s})\|^2] + 8\eta^2\frac{t\sigma^2}{(1-\beta)^2} + 4\eta^2\big(\frac{\beta}{1-\beta}\big)^2 c_m^2 + 2r^2 c_m^2.
\tag{18}
$$

Now we need to bound $\mathbb{E}_{t_0}[\sum_{s=1}^{t-1}\|\nabla f(w_{t_0+s})\|^2]$. By using $\rho$-Lipschitzness of Hessian, we have that

$$
f(w_{t_0+s}) \leq f(w_{t_0+s-1}) - \eta\langle\nabla f(w_{t_0+s-1}), m_{t_0+s-1}\rangle + \frac{1}{2}\eta^2 m_{t_0+s-1}^\top\nabla^2 f(w_{t_0+s-1})m_{t_0+s-1} + \frac{\rho}{6}\eta^3\|m_{t_0+s-1}\|^3.
\tag{19}
$$

By adding $\eta\langle\nabla f(w_{t_0+s-1}), g_{t_0+s-1}\rangle$ on both sides, we have

$$
\begin{aligned}
\eta\langle\nabla f(w_{t_0+s-1}), g_{t_0+s-1}\rangle \leq{}& f(w_{t_0+s-1}) - f(w_{t_0+s}) + \eta\langle\nabla f(w_{t_0+s-1}), g_{t_0+s-1} - m_{t_0+s-1}\rangle \\
&+ \frac{1}{2}\eta^2 m_{t_0+s-1}^\top\nabla^2 f(w_{t_0+s-1})m_{t_0+s-1} + \frac{\rho}{6}\eta^3\|m_{t_0+s-1}\|^3.
\end{aligned}
\tag{20}
$$

Taking conditional expectation on both sides leads to

$$
\mathbb{E}_{t_0+s-1}[\eta\|\nabla f(w_{t_0+s-1})\|^2] \leq \mathbb{E}_{t_0+s-1}[f(w_{t_0+s-1}) - f(w_{t_0+s})] + \eta^2 c_h + \frac{\rho}{6}\eta^3 c_m^3,
\tag{21}
$$

where $\mathbb{E}_{t_0+s-1}[\eta\langle\nabla f(w_{t_0+s-1}), g_{t_0+s-1} - m_{t_0+s-1}\rangle + \frac{1}{2}\eta^2 m_{t_0+s-1}^\top\nabla^2 f(w_{t_0+s-1})m_{t_0+s-1}] \leq \eta^2 c_h$ by the GrACE property. We have that for $t_0 \leq t_0 + s - 1$

$$
\begin{aligned}
\mathbb{E}_{t_0}[\eta\|\nabla f(w_{t_0+s-1})\|^2] &= \mathbb{E}_{t_0}[\mathbb{E}_{t_0+s-1}[\eta\|\nabla f(w_{t_0+s-1})\|^2]] \\
&\overset{(21)}{\leq} \mathbb{E}_{t_0}[\mathbb{E}_{t_0+s-1}[f(w_{t_0+s-1}) - f(w_{t_0+s})]] + \eta^2 c_h + \frac{\rho}{6}\eta^3 c_m^3 \\
&= \mathbb{E}_{t_0}[f(w_{t_0+s-1}) - f(w_{t_0+s})] + \eta^2 c_h + \frac{\rho}{6}\eta^3 c_m^3.
\end{aligned}
\tag{22}
$$

Summing the above inequality from $s = 2, 3, \ldots, t$ leads to

$$
\begin{aligned}
\mathbb{E}_{t_0}[\sum_{s=1}^{t-1}\eta\|\nabla f(w_{t_0+s})\|^2] &\leq \mathbb{E}_{t_0}[f(w_{t_0+1}) - f(w_{t_0+t})] + \eta^2(t-1)c_h + \frac{\rho}{6}\eta^3(t-1)c_m^3 \\
&= \mathbb{E}_{t_0}[f(w_{t_0+1}) - f(w_{t_0}) + f(w_{t_0}) - f(w_{t_0+t})] + \eta^2(t-1)c_h + \frac{\rho}{6}\eta^3(t-1)c_m^3 \\
&\overset{(a)}{\leq} \mathbb{E}_{t_0}[f(w_{t_0+1}) - f(w_{t_0})] + \mathcal{F}_{thred} + \eta^2(t-1)c_h + \frac{\rho}{6}\eta^3(t-1)c_m^3,
\end{aligned}
\tag{23}
$$

where $(a)$ is by the assumption (made for proving by contradiction) that $\mathbb{E}_{t_0}[f(w_{t_0}) - f(w_{t_0+s})] \leq \mathcal{F}_{thred}$ for any $0 \leq s \leq \mathcal{T}_{thred}$. By (21) with $s = 1$ and $\eta = r$, we have

$$
\mathbb{E}_{t_0}[r\|\nabla f(w_{t_0})\|^2] \leq \mathbb{E}_{t_0}[f(w_{t_0}) - f(w_{t_0+1})] + r^2 c_h + \frac{\rho}{6}r^3 c_m^3.
\tag{24}
$$

By (23) and (24), we know that

$$\mathbb{E}_{t_0}[\sum_{s=1}^{t-1} \eta \|\nabla f(w_{t_0+s})\|^2] \le \mathbb{E}_{t_0}[r\|f(w_{t_0})\|^2] + \mathbb{E}_{t_0}[\sum_{s=1}^{t-1} \eta \|\nabla f(w_{t_0+s})\|^2]$$

$$\le \mathcal{F}_{thred} + r^2 c_h + \frac{\rho}{6} r^3 c_m^3 + \eta^2 t c_h + \frac{\rho}{6} \eta^3 t c_m^3$$

$$\overset{(a)}{\le} \mathcal{F}_{thred} + 2r^2 c_h + \frac{\rho}{6} r^3 c_m^3 + \frac{\rho}{6} r^2 \eta c_m^3.$$

$$\overset{(b)}{\le} \mathcal{F}_{thred} + 2r^2 c_h + \frac{\rho}{3} r^3 c_m^3, \tag{25}$$

where $(a)$ is by the constraint that $\eta^2 t \le r^2$ for $0 \le t \le \mathcal{T}_{thred}$ and $(b)$ is by the constraint that $r \ge \eta$. By combining (25) and (18)

$$\mathbb{E}_{t_0}[\|w_{t_0+t} - w_{t_0}\|^2] \le \mathbb{E}_{t_0}[8\eta^2 \frac{t}{(1-\beta)^2} \sum_{s=1}^{t-1} \|\nabla f(w_{t_0+s})\|^2] + 8\eta^2 \frac{t\sigma^2}{(1-\beta)^2} + 4\eta^2 \left(\frac{\beta}{1-\beta}\right)^2 c_m^2 + 2r^2 c_m^2$$

$$\le \frac{8\eta t \left(\mathcal{F}_{thred} + 2r^2 c_h + \frac{\rho}{3} r^3 c_m^3\right)}{(1-\beta)^2} + 8\eta^2 \frac{t\sigma^2}{(1-\beta)^2} + 4\eta^2 \left(\frac{\beta}{1-\beta}\right)^2 c_m^2 + 2r^2 c_m^2.$$

$$\tag{26}$$

$\square$

# D    PROOF OF LEMMA 2 AND LEMMA 3

**Lemma 2** *Denote $t_0$ any time such that $(t_0 \mod \mathcal{T}_{thred}) = 0$. Let us define a quadratic approximation at $w_{t_0}$, $Q(w) := f(w_{t_0}) + \langle w - w_{t_0}, \nabla f(w_{t_0}) \rangle + \frac{1}{2}(w - w_{t_0})^\top H(w - w_{t_0})$, where $H := \nabla^2 f(w_{t_0})$. Also, define $G_s := (I - \eta \sum_{k=1}^{s} \beta^{s-k} H)$ and*

- $q_{v,t-1} := \left( \Pi_{j=1}^{t-1} G_j \right)\left( -rm_{t_0} \right).$

- $q_{m,t-1} := -\sum_{s=1}^{t-1} \left( \Pi_{j=s+1}^{t-1} G_j \right)\beta^s m_{t_0}.$

- $q_{q,t-1} := -\sum_{s=1}^{t-1} \left( \Pi_{j=s+1}^{t-1} G_j \right) \sum_{k=1}^{s} \beta^{s-k}\left( \nabla f(w_{t_0+k}) - \nabla Q(w_{t_0+s}) \right).$

- $q_{w,t-1} := -\sum_{s=1}^{t-1} \left( \Pi_{j=s+1}^{t-1} G_j \right) \sum_{k=1}^{s} \beta^{s-k} \nabla f(w_{t_0}).$

- $q_{\xi,t-1} := -\sum_{s=1}^{t-1} \left( \Pi_{j=s+1}^{t-1} G_j \right) \sum_{k=1}^{s} \beta^{s-k} \xi_{t_0+k}.$

*Then, $w_{t_0+t} - w_{t_0} = q_{v,t-1} + \eta q_{m,t-1} + \eta q_{q,t-1} + \eta q_{w,t-1} + \eta q_{\xi,t-1}$.*

---

**Notations:**
Denote $t_0$ any time such that $(t_0 \mod \mathcal{T}_{thred}) = 0$. Let us define a quadratic approximation at $w_{t_0}$,

$$Q(w) := f(w_{t_0}) + \langle w - w_{t_0}, \nabla f(w_{t_0}) \rangle + \frac{1}{2}(w - w_{t_0})^\top H(w - w_{t_0}), \qquad (27)$$

where $H := \nabla^2 f(w_{t_0})$. Also, we denote

$$G_s := (I - \eta \sum_{k=1}^{s} \beta^{s-k} H)$$

$$v_{m,s} := \beta^s m_{t_0}$$

$$v_{q,s} := \sum_{k=1}^{s} \beta^{s-k}\left( \nabla f(w_{t_0+k}) - \nabla Q(w_{t_0+s}) \right)$$

$$v_{w,s} := \sum_{k=1}^{s} \beta^{s-k} \nabla f(w_{t_0}) \qquad (28)$$

$$v_{\xi,s} := \sum_{k=1}^{s} \beta^{s-k} \xi_{t_0+k}$$

$$\theta_s := \sum_{k=1}^{s} \beta^{s-k}.$$

---

*Proof.* First, we rewrite $m_{t_0+j}$ for any $j \geq 1$ as follows.

$$
\begin{aligned}
m_{t_0+j} &= \beta^j m_{t_0} + \sum_{k=1}^{j} \beta^{j-k} g_{t_0+k} \\
&= \beta^j m_{t_0} + \sum_{k=1}^{j} \beta^{j-k}\left( \nabla f(w_{t_0+k}) + \xi_{t_0+k} \right).
\end{aligned}
\qquad (29)
$$

We have that

$$
\begin{aligned}
w_{t_0+t} - w_{t_0} &= w_{t_0+t-1} - w_{t_0} - \eta m_{t_0+t-1} \\
&\overset{(a)}{=} w_{t_0+t-1} - w_{t_0} - \eta\Big(\beta^{t-1}m_{t_0} + \sum_{k=1}^{t-1}\beta^{t-1-k}\big(\nabla f(w_{t_0+k}) + \xi_{t_0+k}\big)\Big) \\
&\overset{(b)}{=} w_{t_0+t-1} - w_{t_0} - \eta\sum_{k=1}^{t-1}\beta^{t-1-k}\nabla Q(w_{t_0+t-1}) \\
&\quad - \eta\Big(\beta^{t-1}m_{t_0} + \sum_{k=1}^{t-1}\beta^{t-1-k}\big(\nabla f(w_{t_0+k}) - \nabla Q(w_{t_0+t-1}) + \xi_{t_0+k}\big)\Big) \\
&\overset{(c)}{=} w_{t_0+t-1} - w_{t_0} - \eta\sum_{k=1}^{t-1}\beta^{t-1-k}\big(H(w_{t_0+t-1} - w_{t_0}) + \nabla f(w_{t_0})\big) \\
&\quad - \eta\Big(\beta^{t-1}m_{t_0} + \sum_{k=1}^{t-1}\beta^{t-1-k}\big(\nabla f(w_{t_0+k}) - \nabla Q(w_{t_0+t-1}) + \xi_{t_0+k}\big)\Big) \\
&= \big(I - \eta\sum_{k=1}^{t-1}\beta^{t-1-k}H\big)\big(w_{t_0+t-1} - w_{t_0}\big) \\
&\quad - \eta\Big(\beta^{t-1}m_{t_0} + \sum_{k=1}^{t-1}\beta^{t-1-k}\big(\nabla f(w_{t_0+k}) - \nabla Q(w_{t_0+t-1}) + \nabla f(w_{t_0}) + \xi_{t_0+k}\big)\Big),
\end{aligned}
\tag{30}
$$

where $(a)$ is by using (29) with $j = t - 1$, $(b)$ is by subtracting and adding back the same term, and $(c)$ is by $\nabla Q(w_{t_0+t-1}) = \nabla f(w_{t_0}) + H(w_{t_0+t-1} - w_{t_0})$.

To continue, by using the nations in (28), we can rewrite (30) as

$$
w_{t_0+t} - w_{t_0} = G_{t-1}\big(w_{t_0+t-1} - w_{t_0}\big) - \eta\big(v_{m,t-1} + v_{q,t-1} + v_{w,t-1} + v_{\xi,t-1}\big).
\tag{31}
$$

Recursively expanding (31) leads to

$$
\begin{aligned}
w_{t_0+t} - w_{t_0} &= G_{t-1}\big(w_{t_0+t-1} - w_{t_0}\big) - \eta\big(v_{m,t-1} + v_{q,t-1} + v_{w,t-1} + v_{\xi,t-1}\big) \\
&= G_{t-1}\big(G_{t-2}\big(w_{t_0+t-2} - w_{t_0}\big) - \eta\big(v_{m,t-2} + v_{q,t-2} + v_{w,t-2} + v_{\xi,t-2}\big)\big) \\
&\quad - \eta\big(v_{m,t-1} + v_{q,t-1} + v_{w,t-1} + v_{\xi,t-1}\big) \\
&\overset{(a)}{=} \big(\Pi_{j=1}^{t-1}G_j\big)\big(w_{t_0+1} - w_{t_0}\big) - \eta\sum_{s=1}^{t-1}\big(\Pi_{j=s+1}^{t-1}G_j\big)\big(v_{m,s} + v_{q,s} + v_{w,s} + v_{\xi,s}\big), \\
&\overset{(b)}{=} \big(\Pi_{j=1}^{t-1}G_j\big)\big(-rm_{t_0}\big) - \eta\sum_{s=1}^{t-1}\big(\Pi_{j=s+1}^{t-1}G_j\big)\big(v_{m,s} + v_{q,s} + v_{w,s} + v_{\xi,s}\big),
\end{aligned}
\tag{32}
$$

where $(a)$ we use the notation that $\Pi_{j=s}^{t-1}G_j := G_s \times G_{s+1} \times \ldots\ldots G_{t-1}$ and the notation that $\Pi_{j=t}^{t-1}G_j = 1$ and $(b)$ is by the update rule. By using the definitions of $\{q_{\star,t-1}\}$ in the lemma statement, we complete the proof.

$\square$

**Lemma 3** *Following the notations of Lemma 2, we have that*

$$
\mathbb{E}_{t_0}\big[\|w_{t_0+t} - w_{t_0}\|^2\big] \geq \mathbb{E}_{t_0}\big[\|q_{v,t-1}\|^2\big] + 2\eta\mathbb{E}_{t_0}\big[\langle q_{v,t-1}, q_{m,t-1} + q_{q,t-1} + q_{w,t-1} + q_{\xi,t-1}\rangle\big] := C_{lower}
\tag{33}
$$

*Proof.* Following the proof of Lemma 2, we have

$$
w_{t_0+t} - w_{t_0} = q_{v,t-1} + \eta\big(q_{m,t-1} + q_{q,t-1} + q_{w,t-1} + q_{\xi,t-1}\big).
\tag{34}
$$

Therefore, by using $\|a + b\|^2 \geq \|a\|^2 + 2\langle a, b\rangle$,

$$\mathbb{E}_{t_0}[\|w_{t_0+t} - w_{t_0}\|^2] \geq \mathbb{E}_{t_0}[\|q_{v,t-1}\|^2] + 2\eta\mathbb{E}_{t_0}[\langle q_{v,t-1}, q_{m,t-1} + q_{q,t-1} + q_{w,t-1} + q_{\xi,t-1}\rangle]. \tag{35}$$

$\square$

# E   PROOF OF LEMMA 5

**Lemma 5** *Let $\mathcal{F}_{thred} = O(\epsilon^4)$ and $\eta^2 \mathcal{T}_{thred} \leq r^2$. By following the conditions and notations in Theorem 1, Lemma 1 and Lemma 2, we conclude that if SGD with momentum (Algorithm 2) has the APCG property, then we have that $C_{lower} := \mathbb{E}_{t_0}[\|q_{v,\mathcal{T}_{thred}-1}\|^2] + 2\eta \mathbb{E}_{t_0}[\langle q_{v,\mathcal{T}_{thred}-1}, q_{m,\mathcal{T}_{thred}-1} + q_{q,\mathcal{T}_{thred}-1} + q_{w,\mathcal{T}_{thred}-1} + q_{\xi,\mathcal{T}_{thred}-1}\rangle] > C_{upper}$.*

Table 3: Constraints and choices of the parameters.

| Parameter | Value | Constraint origin | constant |
|---|---|---|---|
| $r$ | $\delta\gamma\epsilon^2 c_r$ | (64), (65), (66) | $c_r \leq \frac{c_0}{c_m^3 \rho L \sigma^2 c_h}, c_0 = \frac{1}{1152}$  $\frac{c_0}{c_m^3 \rho L \sigma^2 c'(1-\beta)^2 c_h} \leq c_r$  $\frac{c_0}{c_m^3 \rho L \sigma^4 (1-\beta)^3 c_h} \leq c_r$ |
| $r$ | ” | $r \leq \sqrt{\frac{\delta \mathcal{F}_{thred}}{8c_h}}$ from (89) | ” |
| $\eta$ | $\delta^2 \gamma^2 \epsilon^5 c_\eta$ | (64) | $c_\eta \leq \frac{c_1}{c_m^5 \rho L^2 \sigma^2 c' c_h}, c_1 = \frac{c_0}{24}$ |
| $\eta$ | ” | $\eta \leq r/\sqrt{\mathcal{T}_{thred}}$ from (25),(39),(87),(89) | ” |
| $\eta$ | ” | $\eta \leq \min\{\frac{(1-\beta)}{L}, \frac{(1-\beta)}{\epsilon}\}$ from (45), (78) [7] | ” |
| $\mathcal{F}_{thred}$ | $\delta\gamma^2 \epsilon^4 c_F$ | (65) | $c_F \leq \frac{c_2}{c_m^4 \rho^2 L \sigma^4 c_h}, c_2 = \frac{c_0}{576}$  $c_F \geq \frac{8c_0^2}{c_m^6 \rho^2 L^2 \sigma^4 c_h}$ |
| $\mathcal{F}_{thred}$ | ” | $\mathcal{F}_{thred} \leq \frac{\epsilon^2 r}{4}$ from (88) | ” |
| $\mathcal{T}_{thred}$ | | $\mathcal{T}_{thred} \geq \frac{c(1-\beta)}{\eta\epsilon} \log(\frac{Lc_m \sigma^2 \rho c' c_h}{(1-\beta)\delta\gamma\epsilon})$ from (82) | |

W.l.o.g, we assume that $c_m$, $L$, $\sigma^2$, $c'$, $c_h$, and $\rho$ are not less than one and that $\epsilon \leq 1$.

## E.1   SOME CONSTRAINTS ON $\beta$.

We require that parameter $\beta$ is not too close to 1 so that the following holds,

- 1) $L(1-\beta)^3 > 1$.
- 2) $\sigma^2(1-\beta)^3 > 1$.
- 3) $c'(1-\beta)^2 > 1$.
- 4) $\eta \leq \frac{1-\beta}{L}$.
- 5) $\eta \leq \frac{1-\beta}{\epsilon}$.
- 6) $\mathcal{T}_{thred} \geq \frac{c(1-\beta)}{\eta\epsilon} \log(\frac{Lc_m \sigma^2 \rho c' c_h}{(1-\beta)\delta\gamma\epsilon}) \geq 1 + \frac{2\beta}{1-\beta}$.

The constraints upper-bound the value of $\beta$. That is, $\beta$ cannot be too close to 1. We note that the $\beta$ dependence on $L$, $\sigma$, and $c'$ are only artificial. We use these constraints in our proofs but they are mostly artefacts of the analysis. For example, if a function is $L$-smooth, and $L < 1$, then it is also 1-smooth, so we can assume without loss of generality that $L > 1$. Similarly, the dependence on $\sigma$ is not highly relevant, since we can always increase the variance of the stochastic gradient, for example by adding an $O(1)$ gaussian perturbation.

### E.2 SOME LEMMAS

To prove Lemma 5, we need a series of lemmas with the choices of parameters on Table 3.

**Upper bounding $\mathbb{E}_{t_0}[\|q_{q,t-1}\|]$:**

---

**Lemma 9.** *Following the conditions in Lemma 1 and Lemma 2, we have*

$$
\begin{aligned}
\mathbb{E}_{t_0}[\|q_{q,t-1}\|] \leq & \left(\Pi_{j=1}^{t-1}(1+\eta\theta_j\lambda)\right)\frac{\beta L c_m}{\epsilon(1-\beta)^2} \\
& + \frac{\left(\Pi_{j=1}^{t-1}(1+\eta\theta_j\lambda)\right)}{1-\beta}\frac{\rho}{\eta\epsilon^2}\frac{8\left(\mathcal{F}_{thred}+2r^2c_h+\frac{\rho}{3}r^3c_m^3\right)}{(1-\beta)^2} \\
& + \frac{\left(\Pi_{j=1}^{t-1}(1+\eta\theta_j\lambda)\right)}{1-\beta}\frac{\rho\left(8\frac{r^2\sigma^2}{(1-\beta)^2}+4\eta^2\left(\frac{\beta}{1-\beta}\right)^2c_m^2+2r^2c_m^2\right)}{2\eta\epsilon}.
\end{aligned} \tag{36}
$$

---

*Proof.*

$$
\mathbb{E}_{t_0}[\|q_{q,t-1}\|] = \mathbb{E}_{t_0}[\|-\sum_{s=1}^{t-1}\left(\Pi_{j=s+1}^{t-1}G_j\right)\sum_{k=1}^{s}\beta^{s-k}\left(\nabla f(w_{t_0+k})-\nabla Q(w_{t_0+s})\right)\|]
$$

$$
\overset{(a)}{\leq} \mathbb{E}_{t_0}[\sum_{s=1}^{t-1}\|\left(\Pi_{j=s+1}^{t-1}G_j\right)\sum_{k=1}^{s}\beta^{s-k}\left(\nabla f(w_{t_0+k})-\nabla Q(w_{t_0+s})\right)\|]
$$

$$
\overset{(b)}{\leq} \mathbb{E}_{t_0}[\sum_{s=1}^{t-1}\|\left(\Pi_{j=s+1}^{t-1}G_j\right)\|_2\|\sum_{k=1}^{s}\beta^{s-k}\left(\nabla f(w_{t_0+k})-\nabla Q(w_{t_0+s})\right)\|]
$$

$$
\overset{(c)}{\leq} \mathbb{E}_{t_0}[\sum_{s=1}^{t-1}\|\left(\Pi_{j=s+1}^{t-1}G_j\right)\|_2\sum_{k=1}^{s}\beta^{s-k}\|\left(\nabla f(w_{t_0+k})-\nabla Q(w_{t_0+s})\right)\|]
$$

$$
\overset{(d)}{\leq} \mathbb{E}_{t_0}[\sum_{s=1}^{t-1}\|\left(\Pi_{j=s+1}^{t-1}G_j\right)\|_2\sum_{k=1}^{s}\beta^{s-k}\left(\|\nabla f(w_{t_0+k})-\nabla f(w_{t_0+s})\|+\|\nabla f(w_{t_0+s})-\nabla Q(w_{t_0+s})\|\right)]
$$

$$\tag{37}$$

where $(a)$, $(c)$, $(d)$ is by triangle inequality, $(b)$ is by the fact that $\|Ax\|_2 \leq \|A\|_2\|x\|_2$ for any matrix $A$ and vector $x$. Now that we have an upper bound of $\|\nabla f(w_{t_0+k}) - \nabla f(w_{t_0+s})\|$,

$$
\|\nabla f(w_{t_0+k})-\nabla f(w_{t_0+s})\| \overset{(a)}{\leq} L\|w_{t_0+k}-w_{t_0+s}\| \overset{(b)}{\leq} L\eta(s-k)c_m. \tag{38}
$$

where $(a)$ is by the assumption of L-Lipschitz gradient and $(b)$ is by applying the triangle inequality $(s-k)$ times and that $\|w_t - w_{t-1}\| \leq \eta\|m_{t-1}\| \leq \eta c_m$, for any $t$. We can also derive an upper bound of $\mathbb{E}_{t_0}[\|\nabla f(w_{t_0+s})-\nabla Q(w_{t_0+s})\|]$,

$$
\mathbb{E}_{t_0}[\|\nabla f(w_{t_0+s})-\nabla Q(w_{t_0+s})\|]
$$
$$
\overset{(a)}{\leq} \mathbb{E}_{t_0}[\frac{\rho}{2}\|w_{t_0+s}-w_{t_0}\|^2] \overset{(b)}{\leq} \frac{\rho}{2}\left(\frac{8\eta s\left(\mathcal{F}_{thred}+2r^2c_h+\frac{\rho}{3}r^3c_m^3\right)}{(1-\beta)^2}+8\frac{r^2\sigma^2}{(1-\beta)^2}+4\eta^2\left(\frac{\beta}{1-\beta}\right)^2c_m^2+2r^2c_m^2\cdot\right)
$$
$$\tag{39}$$

Above, $(a)$ is by the fact that if a function $f(\cdot)$ has $\rho$ Lipschitz Hessian, then

$$
\|\nabla f(y)-\nabla f(x)-\nabla^2 f(x)(y-x)\| \leq \frac{\rho}{2}\|y-x\|^2 \tag{40}
$$

(c.f. Lemma 1.2.4 in (Nesterov (2013))) and using the definition that

$$
Q(w) := f(w_{t_0})+\langle w-w_{t_0},\nabla f(w_{t_0})\rangle+\frac{1}{2}(w-w_{t_0})^\top H(w-w_{t_0}),
$$

(b) is by Lemma 1 and $\eta^2 t \leq r^2$ for $0 \leq t \leq \mathcal{T}_{thred}$

$$
\mathbb{E}_{t_0}[\|w_{t_0+t} - w_{t_0}\|^2] \leq \frac{8\eta t\left(\mathcal{F}_{thred} + 2r^2 c_h + \frac{\rho}{3} r^3 c_m^3\right)}{(1-\beta)^2} + 8\eta^2 \frac{t\sigma^2}{(1-\beta)^2} + 4\eta^2\left(\frac{\beta}{1-\beta}\right)^2 c_m^2 + 2r^2 c_m^2
$$

$$
\leq \frac{8\eta t\left(\mathcal{F}_{thred} + 2r^2 c_h + \frac{\rho}{3} r^3 c_m^3\right)}{(1-\beta)^2} + 8\frac{r^2\sigma^2}{(1-\beta)^2} + 4\eta^2\left(\frac{\beta}{1-\beta}\right)^2 c_m^2 + 2r^2 c_m^2.
$$

$$(41)$$

Combing (37), (38), (39), we have that

$$
\mathbb{E}_{t_0}[\|q_{q,t-1}\|]
$$

$$
\overset{(37)}{\leq} \mathbb{E}_{t_0}\Big[\sum_{s=1}^{t-1} \|\big(\Pi_{j=s+1}^{t-1} G_j\big)\|_2 \sum_{k=1}^{s} \beta^{s-k}\big(\|\nabla f(w_{t_0+k}) - \nabla f(w_{t_0+s})\| + \|\nabla f(w_{t_0+s}) - \nabla Q(w_{t_0+s})\|\big)\Big]
$$

$$
\overset{(38),(39)}{\leq} \sum_{s=1}^{t-1} \|\big(\Pi_{j=s+1}^{t-1} G_j\big)\|_2 \sum_{k=1}^{s} \beta^{s-k} L\eta(s-k)c_m
$$

$$
+ \sum_{s=1}^{t-1} \|\big(\Pi_{j=s+1}^{t-1} G_j\big)\|_2 \sum_{k=1}^{s} \beta^{s-k} \frac{\rho}{2}\Big(\frac{8\eta s\left(\mathcal{F}_{thred} + 2r^2 c_h + \frac{\rho}{3} r^3 c_m^3\right)}{(1-\beta)^2} + 8\frac{r^2\sigma^2}{(1-\beta)^2} + 4\eta^2\left(\frac{\beta}{1-\beta}\right)^2 c_m^2 + 2r^2 c_m^2\Big)
$$

$$
:= \sum_{s=1}^{t-1} \|\big(\Pi_{j=s+1}^{t-1} G_j\big)\|_2 \sum_{k=1}^{s} \beta^{s-k} L\eta(s-k)c_m + \sum_{s=1}^{t-1} \|\big(\Pi_{j=s+1}^{t-1} G_j\big)\|_2 \sum_{k=1}^{s} \beta^{s-k} \frac{\rho}{2}(\nu_s + \nu),
$$

$$(42)$$

where on the last line we use the notation that

$$
\nu_s := \frac{8\eta s\left(\mathcal{F}_{thred} + 2r^2 c_h + \frac{\rho}{3} r^3 c_m^3\right)}{(1-\beta)^2}
$$

$$
\nu := 8\frac{r^2\sigma^2}{(1-\beta)^2} + 4\eta^2\left(\frac{\beta}{1-\beta}\right)^2 c_m^2 + 2r^2 c_m^2.
$$

$$(43)$$

To continue, let us analyze $\|\big(\Pi_{j=s+1}^{t-1} G_j\big)\|_2$ first.

$$
\|\big(\Pi_{j=s+1}^{t-1} G_j\big)\|_2 = \|\Pi_{j=s+1}^{t-1}(I - \eta \sum_{k=1}^{j} \beta^{j-k} H)\|_2
$$

$$
\overset{(a)}{\leq} \Pi_{j=s+1}^{t-1}(1 + \eta\theta_j\lambda) = \frac{\Pi_{j=1}^{t-1}(1 + \eta\theta_j\lambda)}{\Pi_{j=1}^{s}(1 + \eta\theta_j\lambda)} \overset{(b)}{\leq} \frac{\Pi_{j=1}^{t-1}(1 + \eta\theta_j\lambda)}{(1 + \eta\epsilon)^s}.
$$

$$(44)$$

Above, we use the notation that $\theta_j := \sum_{k=1}^{j} \beta^{j-k}$. For (a), it is due to that $\lambda := -\lambda_{min}(H)$, $\lambda_{\max}(H) \leq L$, and the choice of $\eta$ so that $1 \geq \frac{\eta L}{1-\beta}$, or equivalently,

$$
\eta \leq \frac{1-\beta}{L}.
$$

$$(45)$$

For (b), it is due to that $\theta_j \geq 1$ for any $j$ and $\lambda \geq \epsilon$. Therefore, we can upper-bound the first term on r.h.s of (42) as

$$
\sum_{s=1}^{t-1} \|\big(\Pi_{j=s+1}^{t-1} G_j\big)\|_2 \sum_{k=1}^{s} \beta^{s-k} L\eta(s-k)c_m = \sum_{s=1}^{t-1} \|\big(\Pi_{j=s+1}^{t-1} G_j\big)\|_2 \sum_{k=1}^{s-1} \beta^k k L\eta c_m
$$

$$
\overset{(a)}{\leq} \sum_{s=1}^{t-1} \|\big(\Pi_{j=s+1}^{t-1} G_j\big)\|_2 \frac{\beta}{(1-\beta)^2} L\eta c_m
$$

$$
\overset{(b)}{\leq} \big(\Pi_{j=1}^{t-1}(1 + \eta\theta_j\lambda)\big)\frac{\beta L\eta c_m}{(1-\beta)^2} \sum_{s=1}^{t-1} \frac{1}{(1 + \eta\epsilon)^s}
$$

$$
\overset{(c)}{\leq} \big(\Pi_{j=1}^{t-1}(1 + \eta\theta_j\lambda)\big)\frac{\beta L\eta c_m}{(1-\beta)^2} \frac{1}{\eta\epsilon} = \big(\Pi_{j=1}^{t-1}(1 + \eta\theta_j\lambda)\big)\frac{\beta L c_m}{\epsilon(1-\beta)^2},
$$

$$(46)$$

where $(a)$ is by that fact that $\sum_{k=1}^{\infty} \beta^k k \leq \frac{\beta}{(1-\beta)^2}$ for any $0 \leq \beta < 1$, $(b)$ is by using (44), and $(c)$ is by using that $\sum_{s=1}^{\infty}(\frac{1}{1+\eta\epsilon})^s \leq \frac{1}{\eta\epsilon}$. Now let us switch to bound $\sum_{s=1}^{t-1}\|(\Pi_{j=s+1}^{t-1}G_j)\|_2\sum_{k=1}^{s}\beta^{s-k}\frac{\rho}{2}(\nu_s+\nu)$ on (42). We have that

$$
\begin{aligned}
\sum_{s=1}^{t-1}&\|(\Pi_{j=s+1}^{t-1}G_j)\|_2\sum_{k=1}^{s}\beta^{s-k}\frac{\rho}{2}(\nu_s+\nu) \overset{(a)}{\leq} \frac{1}{1-\beta}\sum_{s=1}^{t-1}\|(\Pi_{j=s+1}^{t-1}G_j)\|_2\frac{\rho}{2}(\nu_s+\nu)\\
&\overset{(b)}{\leq} \frac{(\Pi_{j=1}^{t-1}(1+\eta\theta_j\lambda))}{1-\beta}\sum_{s=1}^{t-1}\frac{1}{(1+\eta\epsilon)^s}\frac{\rho}{2}\nu_s + \frac{(\Pi_{j=1}^{t-1}(1+\eta\theta_j\lambda))}{1-\beta}\sum_{s=1}^{t-1}\frac{1}{(1+\eta\epsilon)^s}\frac{\rho}{2}\nu\\
&\overset{(c)}{\leq} \frac{(\Pi_{j=1}^{t-1}(1+\eta\theta_j\lambda))}{1-\beta}\sum_{s=1}^{t-1}\frac{1}{(1+\eta\epsilon)^s}\frac{\rho}{2}\nu_s + \frac{(\Pi_{j=1}^{t-1}(1+\eta\theta_j\lambda))}{1-\beta}\frac{\rho\nu}{2\eta\epsilon}\\
&= \frac{(\Pi_{j=1}^{t-1}(1+\eta\theta_j\lambda))}{1-\beta}\sum_{s=1}^{t-1}\frac{1}{(1+\eta\epsilon)^s}\frac{\rho}{2}\nu_s + \frac{(\Pi_{j=1}^{t-1}(1+\eta\theta_j\lambda))}{1-\beta}\frac{\rho\big(8\frac{r^2\sigma^2}{(1-\beta)^2}+4\eta^2\big(\frac{\beta}{1-\beta}\big)^2c_m^2+2r^2c_m^2\big)}{2\eta\epsilon}\\
&\overset{(d)}{\leq} \frac{(\Pi_{j=1}^{t-1}(1+\eta\theta_j\lambda))}{1-\beta}\frac{\rho}{(\eta\epsilon)^2}\frac{8\eta\big(\mathcal{F}_{thred}+2r^2c_h+\frac{\rho}{3}r^3c_m^3\big)}{(1-\beta)^2}\\
&+ \frac{(\Pi_{j=1}^{t-1}(1+\eta\theta_j\lambda))}{1-\beta}\frac{\rho\big(8\frac{r^2\sigma^2}{(1-\beta)^2}+4\eta^2\big(\frac{\beta}{1-\beta}\big)^2c_m^2+2r^2c_m^2\big)}{2\eta\epsilon}
\end{aligned}
\tag{47}
$$

where $(a)$ is by the fact that $\sum_{k=1}^{s}\beta^{s-k} \leq 1/(1-\beta)$, $(b)$ is by (44), $(c)$ is by using that $\sum_{s=1}^{\infty}(\frac{1}{1+\eta\epsilon})^s \leq \frac{1}{\eta\epsilon}$, $(d)$ is by $\sum_{k=1}^{\infty}z^k k \leq \frac{z}{(1-z)^2}$ for any $|z| \leq 1$ and substituting $z = \frac{1}{1+\eta\epsilon}$, which leads to $\sum_{k=1}^{\infty}z^k k \leq \frac{z}{(1-z)^2} = \frac{1/(1+\eta\epsilon)}{(1-1/(1+\eta\epsilon))^2} = \frac{1+\eta\epsilon}{(\eta\epsilon)^2} \leq \frac{2}{(\eta\epsilon)^2}$ in which the last inequality is by chosen the step size $\eta$ so that $\eta\epsilon \leq 1$.

By combining (42), (46), and (47), we have that

$$
\begin{aligned}
\mathbb{E}_{t_0}[\|q_{q,t-1}\|] &\overset{(42)}{\leq} \sum_{s=1}^{t-1}\|(\Pi_{j=s+1}^{t-1}G_j)\|_2\sum_{k=1}^{s}\beta^{s-k}L\eta(s-k)c_m + \sum_{s=1}^{t-1}\|(\Pi_{j=s+1}^{t-1}G_j)\|_2\sum_{k=1}^{s}\beta^{s-k}\frac{\rho}{2}(\nu_s+\nu)\\
&\overset{(46),(47)}{\leq} \big(\Pi_{j=1}^{t-1}(1+\eta\theta_j\lambda)\big)\frac{\beta Lc_m}{\epsilon(1-\beta)^2}\\
&+ \frac{(\Pi_{j=1}^{t-1}(1+\eta\theta_j\lambda))}{1-\beta}\frac{\rho}{\eta\epsilon^2}\frac{8\big(\mathcal{F}_{thred}+2r^2c_h+\frac{\rho}{3}r^3c_m^3\big)}{(1-\beta)^2}\\
&+ \frac{(\Pi_{j=1}^{t-1}(1+\eta\theta_j\lambda))}{1-\beta}\frac{\rho\big(8\frac{r^2\sigma^2}{(1-\beta)^2}+4\eta^2\big(\frac{\beta}{1-\beta}\big)^2c_m^2+2r^2c_m^2\big)}{2\eta\epsilon},
\end{aligned}
\tag{48}
$$

which completes the proof.

$\square$

**Upper bounding** $\|q_{v,t-1}\|$**:**

> **Lemma 10.** *Following the conditions in Lemma 1 and Lemma 2, we have*
> $$\|q_{v,t-1}\| \leq \left(\Pi_{j=1}^{t-1}(1+\eta\theta_j\lambda)\right)rc_m. \tag{49}$$

*Proof.*

$$\|q_{v,t-1}\| \leq \|\left(\Pi_{j=1}^{t-1}G_j\right)\left(-rm_{t_0}\right)\| \leq \|\left(\Pi_{j=1}^{t-1}G_j\right)\|_2\| -rm_{t_0}\| \leq \left(\Pi_{j=1}^{t-1}(1+\eta\theta_j\lambda)\right)rc_m, \tag{50}$$

where the last inequality is because $\eta$ is chosen so that $1 \geq \frac{\eta L}{1-\beta}$ and the fact that $\lambda_{\max}(H) \leq L$.

$\square$

**Lower bounding** $\mathbb{E}_{t_0}[2\eta\langle q_{v,t-1}, q_{q,t-1}\rangle]$**:**

> **Lemma 11.** *Following the conditions in Lemma 1 and Lemma 2, we have*
>
> $\mathbb{E}_{t_0}[2\eta\langle q_{v,t-1}, q_{q,t-1}\rangle]$
> $\geq -2\eta\left(\Pi_{j=1}^{t-1}(1+\eta\theta_j\lambda)\right)^2 rc_m\times$
> $$\left[\frac{\beta Lc_m}{\epsilon(1-\beta)^2} + \frac{\rho}{\eta\epsilon^2}\frac{8\left(\mathcal{F}_{thred} + 2r^2c_h + \frac{\rho}{3}r^3c_m^3\right)}{(1-\beta)^3} + \frac{\rho\left(8\frac{r^2\sigma^2}{(1-\beta)^2} + 4\eta^2\left(\frac{\beta}{1-\beta}\right)^2c_m^2 + 2r^2c_m^2\right)}{2\eta\epsilon(1-\beta)}\right]. \tag{51}$$

*Proof.* By the results of Lemma 9 and Lemma 10

$\mathbb{E}_{t_0}[2\eta\langle q_{v,t-1}, q_{q,t-1}\rangle] \geq -\mathbb{E}_{t_0}[2\eta\|q_{v,t-1}\|\|q_{q,t-1}\|]$

$\overset{Lemma\ 10}{\geq} -\mathbb{E}_{t_0}[2\eta\left(\Pi_{j=1}^{t-1}(1+\eta\theta_j\lambda)\right)rc_m\|q_{q,t-1}\|]$

$\overset{Lemma\ 9}{\geq} -2\eta\left(\Pi_{j=1}^{t-1}(1+\eta\theta_j\lambda)\right)^2 rc_m\times$ $\tag{52}$

$$\left[\frac{\beta Lc_m}{\epsilon(1-\beta)^2} + \frac{\rho}{\eta\epsilon^2}\frac{8\left(\mathcal{F}_{thred} + 2r^2c_h + \frac{\rho}{3}r^3c_m^3\right)}{(1-\beta)^3} + \frac{\rho\left(8\frac{r^2\sigma^2}{(1-\beta)^2} + 4\eta^2\left(\frac{\beta}{1-\beta}\right)^2c_m^2 + 2r^2c_m^2\right)}{2\eta\epsilon(1-\beta)}\right].$$

$\square$

**Lower bounding $\mathbb{E}_{t_0}[2\eta\langle q_{v,t-1}, q_{\xi,t-1}\rangle]$:**

**Lemma 12.** *Following the conditions in Lemma 1 and Lemma 2, we have*
$$\mathbb{E}_{t_0}[2\eta\langle q_{v,t-1}, q_{\xi,t-1}\rangle] = 0. \tag{53}$$

*Proof.*

$$
\begin{aligned}
\mathbb{E}_{t_0}[2\eta\langle q_{v,t-1}, q_{\xi,t-1}\rangle] &= \mathbb{E}_{t_0}[2\eta\langle q_{v,t-1}, -\sum_{s=1}^{t-1}\left(\Pi_{j=s+1}^{t-1}G_j\right)\sum_{k=1}^{s}\beta^{s-k}\xi_{t_0+k}\rangle] \\
&\overset{(a)}{=} \mathbb{E}_{t_0}[2\eta\langle q_{v,t-1}, \sum_{k=1}^{s}\alpha_k\xi_{t_0+k}\rangle] \\
&\overset{(b)}{=} \mathbb{E}_{t_0}[2\eta\sum_{k=1}^{s}\mathbb{E}_{t_0+k-1}[\langle q_{v,t-1}, \alpha_k\xi_{t_0+k}\rangle]] \\
&\overset{(c)}{=} \mathbb{E}_{t_0}[2\eta\sum_{k=1}^{s}\langle q_{v,t-1}, \mathbb{E}_{t_0+k-1}[\alpha_k\xi_{t_0+k}]\rangle] \\
&= \mathbb{E}_{t_0}[2\eta\sum_{k=1}^{s}\alpha_k\langle q_{v,t-1}, \mathbb{E}_{t_0+k-1}[\xi_{t_0+k}]\rangle] \\
&\overset{(d)}{=} 0,
\end{aligned}
\tag{54}
$$

where $(a)$ holds for some coefficients $\alpha_k$, $(b)$ is by the tower rule, $(c)$ is because $q_{v,t-1}$ is measureable with $t_0$, and $(d)$ is by the zero mean assumption of $\xi$'s.

$\square$

**Lower bounding $\mathbb{E}_{t_0}[2\eta\langle q_{v,t-1}, q_{m,t-1}\rangle]$:**

**Lemma 13.** *Following the conditions in Lemma 1 and Lemma 2, we have*
$$\mathbb{E}_{t_0}[2\eta\langle q_{v,t-1}, q_{m,t-1}\rangle] \geq 0. \tag{55}$$

*Proof.*

$$
\begin{aligned}
&\mathbb{E}_{t_0}[2\eta\langle q_{v,t-1}, q_{m,t-1}\rangle] \\
&= 2\eta r\mathbb{E}_{t_0}[\langle\left(\Pi_{j=1}^{t-1}G_j\right)m_{t_0}, \sum_{s=1}^{t-1}\left(\Pi_{j=s+1}^{t-1}G_j\right)\beta^s m_{t_0}\rangle] \\
&\overset{(a)}{=} 2\eta r\mathbb{E}_{t_0}[\langle m_{t_0}, Bm_{t_0}\rangle] \overset{(b)}{\geq} 0,
\end{aligned}
\tag{56}
$$

where $(a)$ is by defining the matrix $B := \left(\Pi_{j=1}^{t-1}G_j\right)^{\top}\left(\sum_{s=1}^{t-1}\left(\Pi_{j=s+1}^{t-1}G_j\right)\beta^s\right)$. For (b), notice that the matrix $B$ is symmetric positive semidefinite. To see that the matrix $B$ is symmetric positive semidefinite, observe that each $G_j := (I - \eta\sum_{k=1}^{j}\beta^{j-k}H)$ can be written in the form of $G_j = UD_jU^{\top}$ for some orthonormal matrix $U$ and a diagonal matrix $D_j$. Therefore, the matrix product $\left(\Pi_{j=1}^{t-1}G_j\right)^{\top}\left(\Pi_{j=s+1}^{t-1}G_j\right) = U(\Pi_{j=1}^{t-1}D_j)(\Pi_{j=s+1}^{t-1}D_j)U^{\top}$ is symmetric positive semidefinite as long as each $G_j$ is. So, $(b)$ is by the property of a matrix being symmetric positive semidefinite.

$\square$

**Lower bounding** $2\eta\mathbb{E}_{t_0}[\langle q_{v,t-1}, q_{w,t-1}\rangle]$**:**

> **Lemma 14.** *Following the conditions in Lemma 1 and Lemma 2, if SGD with momentum has the APCG property, then*
>
> $$2\eta\mathbb{E}_{t_0}[\langle q_{v,t-1}, q_{w,t-1}\rangle] \geq -\frac{2\eta r c'}{(1-\beta)}(\Pi_{j=1}^{t-1}(1+\eta\theta_j\lambda))^2\epsilon. \tag{57}$$

*Proof.* Define $D_s := \Pi_{j=1}^{t-1}G_j\Pi_{j=s+1}^{t-1}G_j$.

$$2\eta\mathbb{E}_{t_0}[\langle q_{v,t-1}, q_{w,t-1}\rangle] = 2\eta\mathbb{E}_{t_0}[\langle (\Pi_{j=1}^{t-1}G_j)(rm_{t_0}), \sum_{s=1}^{t-1}(\Pi_{j=s+1}^{t-1}G_j)\sum_{k=1}^{s}\beta^{s-k}\nabla f(w_{t_0})\rangle]$$

$$= 2\eta\mathbb{E}_{t_0}[\langle rm_{t_0}, \sum_{s=1}^{t-1}(\Pi_{j=1}^{t-1}G_j\Pi_{j=s+1}^{t-1}G_j)\sum_{k=1}^{s}\beta^{s-k}\nabla f(w_{t_0})\rangle]$$

$$= 2\eta r\sum_{s=1}^{t-1}\sum_{k=1}^{s}\beta^{s-k}\mathbb{E}_{t_0}[\langle m_{t_0}, D_s\nabla f(w_{t_0})\rangle]$$

$$\overset{(a)}{\geq} -2\eta^2 r c'\sum_{s=1}^{t-1}\sum_{k=1}^{s}\beta^{s-k}\|D_s\|_2\|\nabla f(w_{t_0})\|^2$$

$$\geq -\frac{2\eta^2 r c'}{1-\beta}\sum_{s=1}^{t-1}\|D_s\|_2\|\nabla f(w_{t_0})\|^2, \tag{58}$$

where $(a)$ is by the APCG property. We also have that

$$\|D_s\|_2 = \|\Pi_{j=1}^{t-1}G_j\Pi_{j=s+1}^{t-1}G_j\|_2 \leq \|\Pi_{j=1}^{t-1}G_j\|_2\|\Pi_{j=s+1}^{t-1}G_j\|_2$$

$$\overset{(a)}{\leq} \|\Pi_{j=1}^{t-1}G_j\|_2\frac{\Pi_{j=1}^{t-1}(1+\eta\theta_j\lambda)}{(1+\eta\epsilon)^s} \overset{(b)}{\leq} \frac{(\Pi_{j=1}^{t-1}(1+\eta\theta_j\lambda))^2}{(1+\eta\epsilon)^s} \tag{59}$$

where (a) and (b) is by (44). Substituting the result back to (58), we get

$$2\eta\mathbb{E}_{t_0}[\langle q_{v,t-1}, q_{w,t-1}\rangle] \geq -\frac{2\eta^2 r c'}{1-\beta}\sum_{s=1}^{t-1}\|D_s\|_2\|\nabla f(w_{t_0})\|^2$$

$$\geq -\frac{2\eta^2 r c'}{1-\beta}\sum_{s=1}^{t-1}\frac{(\Pi_{j=1}^{t-1}(1+\eta\theta_j\lambda))^2}{(1+\eta\epsilon)^s}\|\nabla f(w_{t_0})\|^2 \geq -\frac{2\eta^2 r c'}{(1-\beta)\eta\epsilon}(\Pi_{j=1}^{t-1}(1+\eta\theta_j\lambda))^2\|\nabla f(w_{t_0})\|^2 \tag{60}$$

Using the fact that $\|\nabla f(w_{t_0})\| \leq \epsilon$ completes the proof.

$\square$

### E.3 PROOF OF LEMMA 5

Recall that the strategy is proving by contradiction. Assume that the function value does not decrease at least $\mathcal{F}_{thred}$ in $\mathcal{T}_{thred}$ iterations on expectation. Then, we can get an upper bound of the expected distance $\mathbb{E}_{t_0}[\|w_{t_0+\mathcal{T}_{thred}} - w_{t_0}\|^2] \leq C_{\text{upper}}$ but, by leveraging the negative curvature, we can also show a lower bound of the form $\mathbb{E}_{t_0}[\|w_{t_0+\mathcal{T}_{thred}} - w_{t_0}\|^2] \geq C_{\text{lower}}$. The strategy is showing that the lower bound is larger than the upper bound, which leads to the contradiction and concludes that the function value must decrease at least $\mathcal{F}_{thred}$ in $\mathcal{T}_{thred}$ iterations on expectation. To get the contradiction, according to Lemma 1 and Lemma 3, we need to show that

$$\mathbb{E}_{t_0}[\|q_{v,\mathcal{T}_{thred}-1}\|^2] + 2\eta\mathbb{E}_{t_0}[\langle q_{v,\mathcal{T}_{thred}-1}, q_{m,\mathcal{T}_{thred}-1} + q_{q,\mathcal{T}_{thred}-1} + q_{w,\mathcal{T}_{thred}-1} + q_{\xi,\mathcal{T}_{thred}-1}\rangle] > C_{upper}. \tag{61}$$

Yet, by Lemma 13 and Lemma 12, we have that $\eta\mathbb{E}_{t_0}[\langle q_{v,\mathcal{T}_{thred}-1}, q_{m,\mathcal{T}_{thred}-1}\rangle] \geq 0$ and $\eta\mathbb{E}_{t_0}[\langle q_{v,\mathcal{T}_{thred}-1}, q_{\xi,\mathcal{T}_{thred}-1}\rangle] = 0$. So, it suffices to prove that

$$\mathbb{E}_{t_0}[\|q_{v,\mathcal{T}_{thred}-1}\|^2] + 2\eta\mathbb{E}_{t_0}[\langle q_{v,\mathcal{T}_{thred}-1}, q_{q,\mathcal{T}_{thred}-1} + q_{w,\mathcal{T}_{thred}-1}\rangle] > C_{upper}, \tag{62}$$

and it suffices to show that

- $\frac{1}{4}\mathbb{E}_{t_0}[\|q_{v,\mathcal{T}_{thred}-1}\|^2] + 2\eta\mathbb{E}_{t_0}[\langle q_{v,\mathcal{T}_{thred}-1}, q_{q,\mathcal{T}_{thred}-1}\rangle] \geq 0.$
- $\frac{1}{4}\mathbb{E}_{t_0}[\|q_{v,\mathcal{T}_{thred}-1}\|^2] + 2\eta\mathbb{E}_{t_0}[\langle q_{v,\mathcal{T}_{thred}-1}, q_{w,\mathcal{T}_{thred}-1}\rangle] \geq 0.$
- $\frac{1}{4}\mathbb{E}_{t_0}[\|q_{v,\mathcal{T}_{thred}-1}\|^2] \geq C_{upper}.$

### E.3.1 PROVING THAT $\frac{1}{4}\mathbb{E}_{t_0}[\|q_{v,\mathcal{T}_{thred}-1}\|^2] + 2\eta\mathbb{E}_{t_0}[\langle q_{v,\mathcal{T}_{thred}-1}, q_{q,\mathcal{T}_{thred}-1}\rangle] \geq 0$:

By Lemma 4 and Lemma 11, we have that

$$
\frac{1}{4}\mathbb{E}_{t_0}[\|q_{v,\mathcal{T}_{thred}-1}\|^2] + \mathbb{E}_{t_0}[2\eta\langle q_{v,\mathcal{T}_{thred}-1}, q_{q,\mathcal{T}_{thred}-1}\rangle]
$$
$$
\geq \frac{1}{4}\big(\Pi_{j=1}^{\mathcal{T}_{thred}-1}(1+\eta\theta_j\lambda)\big)^2 r^2\gamma - 2\eta\big(\Pi_{j=1}^{\mathcal{T}_{thred}-1}(1+\eta\theta_j\lambda)\big)^2 rc_m
$$
$$
\times \Big[\frac{\beta L c_m}{\epsilon(1-\beta)^2} + \frac{\rho}{\eta\epsilon^2}\frac{8\big(\mathcal{F}_{thred}+2r^2 c_h + \frac{\rho}{3}r^3 c_m^3\big)}{(1-\beta)^3} + \frac{\rho\big(8\frac{r^2\sigma^2}{(1-\beta)^2}+4\eta^2\big(\frac{\beta}{1-\beta}\big)^2 c_m^2 + 2r^2 c_m^2\big)}{2\eta\epsilon(1-\beta)}\Big].
$$
(63)

To show that the above is nonnegative, it suffices to show that

$$
r^2\gamma \geq \frac{24\eta r\beta L c_m^2}{\epsilon(1-\beta)^2},
$$
(64)

and

$$
r^2\gamma \geq \frac{24\eta r c_m\rho}{(1-\beta)\eta\epsilon^2}\frac{8\big(\mathcal{F}_{thred}+2r^2 c_h + \frac{\rho}{3}r^3 c_m^3\big)}{(1-\beta)^2},
$$
(65)

and

$$
r^2\gamma \geq \frac{24\eta r c_m}{1-\beta}\frac{\rho\big(8\frac{r^2\sigma^2}{(1-\beta)^2}+4\eta^2\big(\frac{\beta}{1-\beta}\big)^2 c_m^2 + 2r^2 c_m^2\big)}{2\eta\epsilon}.
$$
(66)

Now w.l.o.g, we assume that $c_m$, $L$, $\sigma^2$, $c'$, and $\rho$ are not less than one and that $\epsilon \leq 1$. By using the values of parameters on Table 3, we have the following results; a sufficient condition of (64) is that

$$
\frac{c_r}{c_\eta} \geq \frac{24L c_m^2\epsilon^2}{(1-\beta)^2}.
$$
(67)

A sufficient condition of (65) is that

$$
\frac{c_r}{c_F} \geq \frac{576 c_m\rho}{(1-\beta)^3},
$$
(68)

and

$$
1 \geq \frac{1152 c_m\rho c_h c_r}{(1-\beta)^3},
$$
(69)

and

$$
1 \geq \frac{192 c_m^4\rho^2 c_r^2}{(1-\beta)^3}.
$$
(70)

A sufficient condition of (66) is that

$$
1 \geq \frac{96 c_m\rho(\sigma^2+3c_m^2)c_r\epsilon}{(1-\beta)^3},
$$
(71)

and a sufficient condition for the above (71), by the assumption that both $\sigma^2 \geq 1$ and $c_m \geq 1$, is

$$
1 \geq \frac{576 c_m^3\rho\sigma^2 c_r\epsilon}{(1-\beta)^3}.
$$
(72)

Now let us verify if (67), (68), (69), (70), (72) are satisfied. For (67), using the constraint of $c_\eta$ on Table 3, we have that $\frac{1}{c_\eta} \geq \frac{c_m^5\rho L^2\sigma^2 c' c_h}{c_1}$. Using this inequality, it suffices to let $c_r \geq \frac{c_0\epsilon^2}{c_m^3\rho L\sigma^2 c' c_h(1-\beta)^2}$ for getting (67), which holds by using the constraint that $c'(1-\beta)^2 > 1$ and $\epsilon \leq 1$.

For (68), using the constraint of $c_F$ on Table 3, we have that $\frac{1}{c_F} \geq \frac{c_m^4 \rho^2 L \sigma^4 c_h}{c_2}$. Using this inequality, it suffices to let $c_r \geq \frac{c_0}{c_m^3 \rho L \sigma^4 (1-\beta)^3}$, which holds by using the constraint that $\sigma^2 (1-\beta)^3 > 1$. For (69), it needs $\frac{(1-\beta)^3}{1152 c_m \rho c_h} \geq \frac{c_0}{c_m^3 \rho L \sigma^2 c_h} \geq c_r$, which hold by using the constraint that $\sigma^2 (1-\beta)^3 > 1$. For (70), it suffices to let $\frac{(1-\beta)^2}{14 c_m^2 \rho} \geq \frac{c_0}{c_m^3 \rho L \sigma^2 c_h} \geq c_r$ which holds by using the constraint that $\sigma^2 (1-\beta)^3 > 1$. For (72), it suffices to let $\frac{(1-\beta)^3}{576 c_m^3 \rho \sigma^2 \epsilon} \geq \frac{c_0}{c_m^3 \rho L \sigma^2 c_h} \geq c_r$, which holds by using the constraint that $L(1-\beta)^3 > 1$ and $\epsilon \leq 1$. Therefore, by choosing the parameter values as Table 3, we can guarantee that $\frac{1}{4} \mathbb{E}_{t_0}[\|q_{v,\mathcal{T}_{thred}-1}\|^2] + 2\eta \mathbb{E}_{t_0}[\langle q_{v,\mathcal{T}_{thred}-1}, q_{q,\mathcal{T}_{thred}-1}\rangle] \geq 0$.

### E.3.2   PROVING THAT $\frac{1}{4}\mathbb{E}_{t_0}[\|q_{v,\mathcal{T}_{thred}-1}\|^2] + 2\eta \mathbb{E}_{t_0}[\langle q_{v,\mathcal{T}_{thred}-1}, q_{w,\mathcal{T}_{thred}-1}\rangle] \geq 0$:

By Lemma 4 and Lemma 14, we have that

$$
\begin{aligned}
&\frac{1}{4}\mathbb{E}_{t_0}[\|q_{v,\mathcal{T}_{thred}-1}\|^2] + 2\eta \mathbb{E}_{t_0}[\langle q_{v,\mathcal{T}_{thred}-1}, q_{w,\mathcal{T}_{thred}-1}\rangle] \\
&\geq \frac{1}{4}\big(\Pi_{j=1}^{\mathcal{T}_{thred}-1}(1+\eta\theta_j\lambda)\big)^2 r^2 \gamma - \frac{2\eta r c'}{(1-\beta)}\big(\Pi_{j=1}^{\mathcal{T}_{thred}-1}(1+\eta\theta_j\lambda)\big)^2 \epsilon.
\end{aligned}
\tag{73}
$$

To show that the above is nonnegative, it suffices to show that

$$
r^2 \gamma \geq \frac{8\eta r c' \epsilon}{(1-\beta)}.
\tag{74}
$$

A sufficient condition is $\frac{c_r}{c_\eta} \geq \frac{8\epsilon^4 c'}{1-\beta}$. Using the constraint of $c_\eta$ on Table 3, we have that $\frac{1}{c_\eta} \geq \frac{c_m^5 \rho L^2 \sigma^2 c' c_h}{c_1}$. So, it suffices to let $c_r \geq \frac{c_0 \epsilon^4}{3 c_m^5 \rho L^2 \sigma^2 c_h (1-\beta)}$, which holds by using the constraint that $L(1-\beta)^3 > 1$ (so that $L(1-\beta) > 1$) and $\epsilon \leq 1$.

### E.3.3   PROVING THAT $\frac{1}{4}\mathbb{E}_{t_0}[\|q_{v,\mathcal{T}_{thred}-1}\|^2] \geq C_{upper}$:

From Lemma 4 and Lemma 1, we need to show that

$$
\begin{aligned}
&\frac{1}{4}\big(\Pi_{j=1}^{\mathcal{T}_{thred}-1}(1+\eta\theta_j\lambda)\big)^2 r^2 \gamma \\
&\geq \frac{8\eta t\big(\mathcal{F}_{thred} + 2r^2 c_h + \frac{\rho}{3}r^3 c_m^3\big)}{(1-\beta)^2} + 8\frac{r^2 \sigma^2}{(1-\beta)^2} + 4\eta^2 \big(\frac{\beta}{1-\beta}\big)^2 c_m^2 + 2r^2 c_m^2.
\end{aligned}
\tag{75}
$$

We know that $\frac{1}{4}\big(\Pi_{j=1}^{\mathcal{T}_{thred}-1}(1+\eta\theta_j\lambda)\big)^2 r^2 \gamma \geq \frac{1}{4}\big(\Pi_{j=1}^{\mathcal{T}_{thred}-1}(1+\eta\theta_j\epsilon)\big)^2 r^2 \gamma$. It suffices to show that

$$
\begin{aligned}
&\frac{1}{4}\big(\Pi_{j=1}^{\mathcal{T}_{thred}-1}(1+\eta\theta_j\epsilon)\big)^2 r^2 \gamma \\
&\geq \frac{8\eta t\big(\mathcal{F}_{thred} + 2r^2 c_h + \frac{\rho}{3}r^3 c_m^3\big)}{(1-\beta)^2} + 8\frac{r^2 \sigma^2}{(1-\beta)^2} + 4\eta^2 \big(\frac{\beta}{1-\beta}\big)^2 c_m^2 + 2r^2 c_m^2.
\end{aligned}
\tag{76}
$$

Note that the left hand side is exponentially growing in $\mathcal{T}_{thred}$. We can choose the number of iterations $\mathcal{T}_{thred}$ large enough to get the desired result. Specifically, we claim that $\mathcal{T}_{thred} \geq \frac{c(1-\beta)}{\eta\epsilon}\log\big(\frac{L c_m \sigma^2 \rho c' c_h}{(1-\beta)\delta\gamma\epsilon}\big)$ for some constant $c > 0$. To see this, let us first apply $\log$ on both sides of (76),

$$
2\Big(\sum_{j=1}^{\mathcal{T}_{thred}-1} \log(1+\eta\theta_j\epsilon)\Big) + \log(r^2\gamma) \geq \log(8a\mathcal{T}_{thred} + 8b)
\tag{77}
$$

where we denote $a := \frac{4\eta\big(\mathcal{F}_{thred} + 2r^2 c_h + \frac{\rho}{3}r^3 c_m^3\big)}{(1-\beta)^2}$ and $b := 4\frac{r^2\sigma^2}{(1-\beta)^2} + 2\eta^2 \big(\frac{\beta}{1-\beta}\big)^2 c_m^2 + r^2 c_m^2$. To proceed, we are going to use the inequality $\log(1+x) \geq \frac{x}{2}$, for $x \in [0, \sim 2.51]$. We have that

$$
1 \geq \frac{\eta\epsilon}{(1-\beta)}
\tag{78}
$$

as guaranteed by the constraint of $\eta$. So,

$$2\Big(\sum_{j=1}^{\mathcal{T}_{thred}-1} \log(1+\eta\theta_j\epsilon)\Big) \overset{(a)}{\geq} \sum_{j=1}^{\mathcal{T}_{thred}-1} \eta\theta_j\epsilon = \sum_{j=1}^{\mathcal{T}_{thred}-1}\sum_{k=0}^{j-1} \beta^k\eta\epsilon$$

$$= \sum_{j=1}^{\mathcal{T}_{thred}-1} \frac{1-\beta^j}{1-\beta}\eta\epsilon \geq \frac{1}{1-\beta}(\mathcal{T}_{thred}-1-\frac{\beta}{1-\beta})\eta\epsilon.$$

$$\overset{(b)}{\geq} \frac{\mathcal{T}_{thred}-1}{2(1-\beta)}\eta\epsilon, \tag{79}$$

where $(a)$ is by using the inequality $\log(1+x) \geq \frac{x}{2}$ with $x = \eta\theta_j\epsilon \leq 1$ and $(b)$ is by making $\frac{\mathcal{T}_{thred}-1}{2(1-\beta)} \geq \frac{\beta}{(1-\beta)^2}$, which is equivalent to the condition that

$$\mathcal{T}_{thred} \geq 1 + \frac{2\beta}{1-\beta} \tag{80}$$

Now let us substitute the result of (79) back to (77). We have that

$$\mathcal{T}_{thred} \geq 1 + \frac{2(1-\beta)}{\eta\epsilon}\log(\frac{8a\mathcal{T}_{thred}+8b}{\gamma r^2}), \tag{81}$$

which is what we need to show. By choosing $\mathcal{T}_{thred}$ large enough,

$$\mathcal{T}_{thred} \geq \frac{c(1-\beta)}{\eta\epsilon}\log(\frac{Lc_m\sigma^2\rho c'c_h}{(1-\beta)\delta\gamma\epsilon}) = O((1-\beta)\log(\frac{1}{(1-\beta)\epsilon})\epsilon^{-6}) \tag{82}$$

for some constant $c > 0$, we can guarantee that the above inequality (81) holds.

## F    PROOF OF LEMMA 15

**Lemma 15** (Daneshmand et al. (2018))  *Let us define the event $\Upsilon_k := \{\|\nabla f(w_{k\mathcal{T}_{thred}})\| \geq \epsilon$ or $\lambda_{\min}(\nabla^2 f(w_{k\mathcal{T}_{thred}})) \leq -\epsilon\}$. The complement is $\Upsilon_k^c := \{\|\nabla f(w_{k\mathcal{T}_{thred}})\| \leq \epsilon$ and $\lambda_{\min}(\nabla^2 f(w_{k\mathcal{T}_{thred}})) \geq -\epsilon\}$, which suggests that $w_{k\mathcal{T}_{thred}}$ is an $(\epsilon, \epsilon)$-second order stationary points. Suppose that*

$$\mathbb{E}[f(w_{(k+1)\mathcal{T}_{thred}}) - f(w_{k\mathcal{T}_{thred}})|\Upsilon_k] \leq -\Delta$$

$$\mathbb{E}[f(w_{(k+1)\mathcal{T}_{thred}}) - f(w_{k\mathcal{T}_{thred}})|\Upsilon_k^c] \leq \delta\frac{\Delta}{2}. \tag{83}$$

*Set $T = 2\mathcal{T}_{thred}(f(w_0) - \min_w f(w))/(\delta\Delta)$.  We return $w$ uniformly randomly from $w_0, w_{\mathcal{T}_{thred}}, w_{2\mathcal{T}_{thred}}, \dots, w_{k\mathcal{T}_{thred}}, \dots, w_{K\mathcal{T}_{thred}}$, where $K := \lfloor T/\mathcal{T}_{thred}\rfloor$. Then, with probability at least $1 - \delta$, we will have chosen a $w_k$ where $\Upsilon_k$ did not occur.*

*Proof.*  Let $P_k$ be the probability that $\Upsilon_k$ occurs.

$$\mathbb{E}[f(w_{(k+1)\mathcal{T}_{thred}}) - f(w_{k\mathcal{T}_{thred}})]$$
$$= \mathbb{E}[f(w_{(k+1)\mathcal{T}_{thred}}) - f(w_{k\mathcal{T}_{thred}})|\Upsilon_k]P_k + \mathbb{E}[f(w_{(k+1)\mathcal{T}_{thred}}) - f(w_{k\mathcal{T}_{thred}})|\Upsilon_k^c](1-P_k)$$
$$\leq -\Delta P_k + \delta\Delta/2(1-P_k)$$
$$= \delta\Delta/2 - (1+\delta/2)\Delta P_k$$
$$\leq \delta\Delta/2 - \Delta P_k. \tag{84}$$

Summing over all $K$, we have

$$\frac{1}{K+1}\sum_{k=0}^{K}\mathbb{E}[f(w_{(k+1)\mathcal{T}_{thred}}) - f(w_{k\mathcal{T}_{thred}})] \leq \Delta\frac{1}{K+1}\sum_{k=0}^{K}(\delta/2 - P_k)$$

$$\Rightarrow \frac{1}{K+1}\sum_{k=0}^{K}P_k \leq \delta/2 + \frac{f(w_0) - \min_w f(w)}{(K+1)\Delta} \leq \delta \tag{85}$$

$$\Rightarrow \frac{1}{K+1}\sum_{k=0}^{K}(1-P_k) \geq 1-\delta.$$

$\square$

# G   PROOF OF THEOREM 1

**Theorem 1** *Assume that the stochastic momentum satisfies CNC. Set $r = O(\epsilon^2)$, $\eta = O(\epsilon^5)$, and $\mathcal{T}_{thred} = \frac{c(1-\beta)}{\eta\epsilon} \log(\frac{Lc_m\sigma^2\rho c' c_h}{(1-\beta)\delta\gamma\epsilon}) = O((1-\beta)\log(\frac{Lc_m\sigma^2\rho c' c_h}{(1-\beta)\delta\gamma\epsilon})\epsilon^{-6})$ for some constant $c > 0$. If SGD with momentum (Algorithm 2) has APAG property when gradient is large ($\|\nabla f(w)\| \geq \epsilon$), $APCG_{\mathcal{T}_{thred}}$ property when it enters a region of saddle points that exhibits a negative curvature ($\|\nabla f(w)\| \leq \epsilon$ and $\lambda_{\min}(\nabla^2 f(w)) \leq -\epsilon$), and GrACE property throughout the iterations, then it reaches an $(\epsilon, \epsilon)$ second order stationary point in $T = 2\mathcal{T}_{thred}(f(w_0) - \min_w f(w))/(\delta\mathcal{F}_{thred}) = O((1-\beta)\log(\frac{Lc_m\sigma^2\rho c' c_h}{(1-\beta)\delta\gamma\epsilon})\epsilon^{-10})$ iterations with high probability $1 - \delta$, where $\mathcal{F}_{thred} = O(\epsilon^4)$.*

## G.1   PROOF SKETCH OF THEOREM 1

In this subsection, we provide a sketch of the proof of Theorem 1. The complete proof is available in Appendix G. Our proof uses a lemma in (Daneshmand et al. (2018)), which is Lemma 15 below. The lemma guarantees that uniformly sampling a $w$ from $\{w_{k\mathcal{T}_{thred}}\}$, $k = 0, 1, 2, \ldots, \lfloor T/\mathcal{T}_{thred} \rfloor$ gives an $(\epsilon, \epsilon)$-second order stationary point with high probability. We replicate the proof of Lemma 15 in Appendix F.

**Lemma 15.** *(Daneshmand et al. (2018)) Let us define the event $\Upsilon_k := \{\|\nabla f(w_{k\mathcal{T}_{thred}})\| \geq \epsilon$ or $\lambda_{\min}(\nabla^2 f(w_{k\mathcal{T}_{thred}})) \leq -\epsilon\}$. The complement is $\Upsilon_k^c := \{\|\nabla f(w_{k\mathcal{T}_{thred}})\| \leq \epsilon$ and $\lambda_{\min}(\nabla^2 f(w_{k\mathcal{T}_{thred}})) \geq -\epsilon\}$, which suggests that $w_{k\mathcal{T}_{thred}}$ is an $(\epsilon, \epsilon)$-second order stationary points. Suppose that*

$$\mathbb{E}[f(w_{(k+1)\mathcal{T}_{thred}}) - f(w_{k\mathcal{T}_{thred}})|\Upsilon_k] \leq -\Delta \quad \& \quad \mathbb{E}[f(w_{(k+1)\mathcal{T}_{thred}}) - f(w_{k\mathcal{T}_{thred}})|\Upsilon_k^c] \leq \delta\frac{\Delta}{2}.$$
(86)

*Set $T = 2\mathcal{T}_{thred}(f(w_0) - \min_w f(w))/(\delta\Delta)$. [8] We return $w$ uniformly randomly from $w_0, w_{\mathcal{T}_{thred}}, w_{2\mathcal{T}_{thred}}, \ldots, w_{k\mathcal{T}_{thred}}, \ldots, w_{K\mathcal{T}_{thred}}$, where $K := \lfloor T/\mathcal{T}_{thred} \rfloor$. Then, with probability at least $1 - \delta$, we will have chosen a $w_k$ where $\Upsilon_k$ did not occur.*

To use the result of Lemma 15, we need to let the conditions in (86) be satisfied. We can bound $\mathbb{E}[f(w_{(k+1)\mathcal{T}_{thred}}) - f(w_{k\mathcal{T}_{thred}})|\Upsilon_k] \leq -\mathcal{F}_{thred}$, based on the analysis of the large gradient norm regime (Lemma 7) and the analysis for the scenario when the update is with small gradient norm but a large negative curvature is available (Subsection 3.2.1). For the other condition, $\mathbb{E}[f(w_{(k+1)\mathcal{T}_{thred}}) - f(w_{k\mathcal{T}_{thred}})|\Upsilon_k^c] \leq \delta\frac{\mathcal{F}_{thred}}{2}$, it requires that the expected amortized increase of function value due to taking the large step size $r$ is limited (i.e. bounded by $\delta\frac{\mathcal{F}_{thred}}{2}$) when $w_{k\mathcal{T}_{thred}}$ is a second order stationary point. By having the conditions satisfied, we can apply Lemma 15 and finish the proof of the theorem.

## G.2   FULL PROOF OF THEOREM 1

*Proof.* Our proof is based on Lemma 15. So, let us consider the events in Lemma 15, $\Upsilon_k := \{\|\nabla f(w_{k\mathcal{T}_{thred}})\| \geq \epsilon$ or $\lambda_{\min}(\nabla^2 f(w_{k\mathcal{T}_{thred}})) \leq -\epsilon\}$. We first show that $\mathbb{E}[f(w_{(k+1)\mathcal{T}_{thred}}) - f(w_{k\mathcal{T}_{thred}})|\Upsilon_k] \leq \mathcal{F}_{thred}$.

**When $\|\nabla f(w_{k\mathcal{T}_{thred}})\| \geq \epsilon$:**

---

[8]One can use any upper bound of $f(w_0) - \min_w f(w)$ as $f(w_0) - \min_w f(w)$ in the expression of $T$.

Consider that $\Upsilon_k$ is the case that $\|\nabla f(w_{k\mathcal{T}_{thred}})\| \geq \epsilon$. Denote $t_0 := k\mathcal{T}_{thred}$ in the following. We have that

$$
\begin{aligned}
\mathbb{E}_{t_0}[f(w_{t_0+\mathcal{T}_{thred}}) - f(w_{t_0})] &= \sum_{t=0}^{\mathcal{T}_{thred}-1} \mathbb{E}_{t_0}[\mathbb{E}[f(w_{t_0+t+1}) - f(w_{t_0+t})|w_{0:t_0+t}]] \\
&= \mathbb{E}_{t_0}[f(w_{t_0+1}) - f(w_{t_0})] + \sum_{t=1}^{\mathcal{T}_{thred}-1} \mathbb{E}_{t_0}[\mathbb{E}[f(w_{t_0+t+1}) - f(w_{t_0+t})|w_{0:t_0+t}]] \\
&\overset{(a)}{\leq} -\frac{r}{2}\|\nabla f(w_{t_0})\|^2 + \frac{Lr^2c_m^2}{2} + \sum_{t=1}^{\mathcal{T}_{thred}-1} \mathbb{E}_{t_0}[\mathbb{E}[f(w_{t_0+t+1}) - f(w_{t_0+t})|w_{0:t_0+t}]] \\
&\overset{(b)}{\leq} -\frac{r}{2}\|\nabla f(w_{t_0})\|^2 + \frac{Lr^2c_m^2}{2} + \sum_{t=1}^{\mathcal{T}_{thred}-1} \left(\eta^2 c_h + \frac{\rho}{6}\eta^3 c_m^3\right) \\
&\overset{(c)}{\leq} -\frac{r}{2}\|\nabla f(w_{t_0})\|^2 + \frac{Lr^2c_m^2}{2} + r^2 c_h + \frac{\rho}{6}r^3 c_m^3 \\
&\overset{(d)}{\leq} -\frac{r}{2}\|\nabla f(w_{t_0})\|^2 + Lr^2c_m^2 + r^2 c_h \\
&\overset{(e)}{\leq} -\frac{r}{2}\epsilon^2 + Lr^2c_m^2 + r^2 c_h \overset{(f)}{\leq} -\frac{r}{4}\epsilon^2 \overset{(g)}{\leq} -\mathcal{F}_{thred},
\end{aligned}
\tag{87}
$$

where $(a)$ is by using Lemma 6 with step size $r$, $(b)$ is by using Lemma 8, $(c)$ is due to the constraint that $\eta^2 \mathcal{T}_{thred} \leq r^2$, $(d)$ is by the choice of $r$, $(e)$ is by $\|\nabla f(w_t)\| \geq \epsilon$, $(f)$ is by the choice of $r$ so that $r \leq \frac{\epsilon^2}{4(Lc_m^2 + c_h)}$, and $(g)$ is by

$$
\frac{r}{4}\epsilon^2 \geq \mathcal{F}_{thred}.
\tag{88}
$$

**When $\|\nabla f(w_{k\mathcal{T}_{thred}})\| \leq \epsilon$ and $\lambda_{\min}(\nabla^2 f(w_{k\mathcal{T}_{thred}})) \leq -\epsilon$:**

The scenario that $\Upsilon_k$ is the case that $\|\nabla f(w_{k\mathcal{T}_{thred}})\| \leq \epsilon$ and $\lambda_{\min}(\nabla^2 f(w_{k\mathcal{T}_{thred}})) \leq -\epsilon$ has been analyzed in Appendix E, which guarantees that $\mathbb{E}[f(w_{t_0+\mathcal{T}_{thred}}) - f(w_{t_0})] \leq -\mathcal{F}_{thred}$ under the setting.

**When $\|\nabla f(w_{k\mathcal{T}_{thred}})\| \leq \epsilon$ and $\lambda_{\min}(\nabla^2 f(w_{k\mathcal{T}_{thred}})) \geq -\epsilon$:**

Now let us switch to show that $\mathbb{E}[f(w_{(k+1)\mathcal{T}_{thred}}) - f(w_{k\mathcal{T}_{thred}})|\Upsilon_k^c] \leq \delta\frac{\mathcal{F}_{thred}}{2}$. Recall that $\Upsilon_k^c$ means that $\|\nabla f(w_{k\mathcal{T}_{thred}})\| \leq \epsilon$ and $\lambda_{\min}(\nabla^2 f(w_{k\mathcal{T}_{thred}})) \geq -\epsilon$. Denote $t_0 := k\mathcal{T}_{thred}$ in the following. We have that

$$
\begin{aligned}
\mathbb{E}_{t_0}[f(w_{t_0+\mathcal{T}_{thred}}) - f(w_{t_0})] &= \sum_{t=0}^{\mathcal{T}_{thred}-1} \mathbb{E}_{t_0}[\mathbb{E}[f(w_{t_0+t+1}) - f(w_{t_0+t})|w_{0:t_0+t}]] \\
&= \mathbb{E}_{t_0}[f(w_{t_0+1}) - f(w_{t_0})] + \sum_{t=1}^{\mathcal{T}_{thred}-1} \mathbb{E}_{t_0}[\mathbb{E}[f(w_{t_0+t+1}) - f(w_{t_0+t})|w_{0:t_0+t}]] \\
&\overset{(a)}{\leq} r^2 c_h + \frac{\rho}{6}r^3 c_m^3 + \sum_{t=1}^{\mathcal{T}_{thred}-1} \mathbb{E}_{t_0}[\mathbb{E}[f(w_{t_0+t+1}) - f(w_{t_0+t})|w_{0:t_0+t}]] \\
&\overset{(b)}{\leq} r^2 c_h + \frac{\rho}{6}r^3 c_m^3 + \sum_{t=1}^{\mathcal{T}_{thred}-1} \left(\eta^2 c_h + \frac{\rho}{6}\eta^3 c_m^3\right) \\
&\overset{(c)}{\leq} 2r^2 c_h + \frac{\rho}{3}r^3 c_m^3 \leq 4r^2 c_h \overset{(d)}{\leq} \frac{\delta\mathcal{F}_{thred}}{2}.
\end{aligned}
\tag{89}
$$

where $(a)$ is by using Lemma 8 with step size $r$, $(b)$ is by using Lemma 8 with step step size $\eta$, $(c)$ is by setting $\eta^2 \mathcal{T}_{thred} \leq r^2$ and $\eta \leq r$, $(d)$ is by the choice of $r$ so that $8r^2 c_h \leq \delta\mathcal{F}_{thred}$.

Now we are ready to use Lemma 15, since both the conditions are satisfied. According to the lemma and the choices of parameters value on Table 3, we can set $T = 2\mathcal{T}_{thred}\big(f(w_0) - \min_w f(w)\big)/(\delta\mathcal{F}_{thred}) = O((1 - \beta)\log(\frac{Lc_m\sigma^2\rho c'c_h}{(1-\beta)\delta\gamma\epsilon})\epsilon^{-10})$, which will return a $w$ that is an $(\epsilon, \epsilon)$ second order stationary point. Thus, we have completed the proof.

$\square$

### G.3   COMPARISON TO DANESHMAND ET AL. (2018)

Theorem 2 in Daneshmand et al. (2018) states that, for CNC-SGD to find an $(\epsilon, \rho^{1/2}\epsilon)$ stationary point, the total number of iterations is $T = O(\frac{\ell^{10}L^3}{\delta^4\gamma^4}\log^2(\frac{\ell L}{\epsilon\delta\gamma})\epsilon^{-10})$, where $\ell$ is the bound of the stochastic gradient norm $\|g_t\| \leq \ell$ which can be viewed as the counterpart of $c_m$ in our paper. By translating their result for finding an $(\epsilon, \epsilon)$ stationary point, it is $T = O(\frac{\ell^{10}L^3\rho^5}{\delta^4\gamma^4}\log^2(\frac{\rho\ell L}{\epsilon\delta\gamma})\epsilon^{-10})$. On the other hand, using the parameters value on Table 3, we have that $T = 2\mathcal{T}_{thred}\big(f(w_0) - \min_w f(w)\big)/(\delta\mathcal{F}_{thred}) = O(\frac{(1-\beta)c_m^9 L^3\rho^3(\sigma^2)^3 c_h^2 c'}{\delta^4\gamma^4}\log(\frac{Lc_m\sigma^2 c'c_h}{(1-\beta)\delta\gamma\epsilon})\epsilon^{-10})$ for Algorithm 2.

Before making a comparison, we note that their result does not have a dependency on the variance of stochastic gradient (i.e. $\sigma^2$), which is because they assume that the variance is also bounded by the constant $\ell$ (can be seen from (86) in the supplementary of their paper where the variance terms $\|\zeta_i\|$ are bounded by $\ell$). Following their treatment, if we assume that $\sigma^2 \leq c_m$, then on (71) we can instead replace $(\sigma^2 + 3c_m^2)$ with $4c_m^2$ and on (72) it becomes $1 \geq \frac{576c_m^3\rho c_r\epsilon}{(1-\beta)^3}$. This will remove all the parameters' dependency on $\sigma^2$. Now by comparing $\tilde{O}((1 - \beta)c_m^9 c_h^2 c' \cdot \frac{\rho^3 L^3}{\delta^4\gamma^4}\epsilon^{-10})$ of ours and $T = \tilde{O}(\rho^2\ell^{10} \cdot \frac{\rho^3 L^3}{\delta^4\gamma^4}\epsilon^{-10})$ of Daneshmand et al. (2018), we see that in the high momentum regime where $(1 - \beta) << \frac{\rho^2\ell^{10}}{c_m^9 c_h^2 c'}$, Algorithm 2 is strictly better than that of Daneshmand et al. (2018), which means that a higher momentum can help to find a second order stationary point faster.

