# OpenReview forum: "Escaping Saddle Points Faster with Stochastic Momentum"
_ICLR.cc/2020/Conference — Accept (Poster)_

### Official Review · AnonReviewer3 · 2019-10-23
**Official Blind Review #3**

**Rating:** 6

**Review:**

Summary:
The paper presents an analysis and numerical evaluation of SGD with momentum for non-convex optimization problems. In particular, the main contribution of the paper, as the authors claim in the abstract, is showing that stochastic momentum improves deep network training because it modifies SGD to escape saddle points faster.

Comments:
SGD with heavy ball momentum (or stochastic heavy ball method) is one of the most popular methods for training neural networks. As the authors mentioned the method is widely employed in practice, especially in the area of deep learning. However, despite the popularity of the method both in convex and non-convex optimization, its convergence properties are not very well understood.
I consider any meaningful direction for understanding the convergence of this method a nice fit to ICLR, however i find the presentation of this paper somehow confusing, especially Section 3 which is supposed to be the main contribution of the paper.

1) I understand the motivation of the authors and what they tried to communicate but i find that there
is no satisfactory explanation of what Theorem 1 is actually saying. For example In case that $\beta=0$ the method is the popular stochastic gradient descent method. Does the theorem covers the known results of SGD appeared in previous works?

2) In addition, the theorem states" If SGD with momentum (Algorithm 2) has ....APAG...APCG_T...GrACE ...". How we can guarantee that the above 3 conditions is satisfied from the Algorithm 2? There is also no satisfactory explanation of why the authors analyze Algorithm 2 and not Algorithm 1.

3) The presentation of Section 3.2.1 is also not clear. I am suggesting to the authors to explain in more details the theoretical results of their paper and highlight why the 5 lemmas of this section are important to be in the main part of the paper. I think a notation subsection will be useful for the reader.

4) I am suggesting the authors to include in the introduction the more known variant of SGD with momentum:
$$\omega_{t+1}=\omega_t-\eta \nabla f(\omega_t, \xi_t)+\beta(\omega_t-\omega_{t-1})$$
5) The authors name their methods "SGD with stochastic momentum". The method is either "SGD with momentum" or "Stochastic heavy ball method". SGD with stochastic momentum is something different (see for example [Loizou, Richtarik 2017] from paper's reference section)

Minor Comment:
page 3, bellow eq(4): the first (3) ----> Problem (3)

Missing references:
The authors did an excellent work on reviewing the literature review on SGD with momentum. Bellow find three recent related works that could be mentioned:

Aybat, Necdet Serhat, Alireza Fallah, Mert Gurbuzbalaban, and Asuman Ozdaglar. "A universally optimal multistage accelerated stochastic gradient method." arXiv preprint arXiv:1901.08022 (2019).
Loizou, Nicolas, and Peter Richtárik. "Linearly convergent stochastic heavy ball method for minimizing generalization error." arXiv preprint arXiv:1710.10737 (2017).
Loizou, Nicolas, and Peter Richtárik. "Accelerated gossip via stochastic heavy ball method." In 2018 56th Annual Allerton Conference on Communication, Control, and Computing (Allerton), pp. 927-934. IEEE, 2018.

========= after rebuttal =============
 I would like to thank the authors for the reply. After reading their response and the comments of the other reviewers I  decide to update my score to "weak accept".


**Experience Assessment:**

I have published in this field for several years.

**Review Assessment: Checking Correctness Of Derivations And Theory:**

I assessed the sensibility of the derivations and theory.

**Review Assessment: Checking Correctness Of Experiments:**

I assessed the sensibility of the experiments.

**Review Assessment: Thoroughness In Paper Reading:**

I read the paper at least twice and used my best judgement in assessing the paper.

---

> ### Author Response · Authors · 2019-11-11
> **Response to comment 1**
>
> We thank you for the comments. We sincerely hope the following response clarifies your concerns.
>
> === Response to comment 1 ===
>
> First, we want to emphasize that all the works of escaping saddle points so far consider some variants of the standard algorithms, instead of directly analyzing the standard ones. The algorithms they considered all have some deviations from the standard ones (as we mentioned in the Section 2.3). That is, we are not aware of any works showing escaping saddle points of the popular/standard SGD. There seems to be some technical challenge. Because of the challenge, we consider occasionally using a large step size, which leads to Algorithm 2 (instead of Algorithm 1). The tweak was also adopted in (Daneshmand et al. 2018) for analyzing (a variant of) SGD.
>
> Second, when $\beta=0$, Algorithm 2 reduces to the algorithm of (Daneshmand et al. 2018). It reduces to a variant of SGD in (Daneshmand et al. 2018) which occasionally boosts the step size, instead of reducing to the popular SGD. Consequently, our theoretical results reduce to the results in (Daneshmand et al. 2018). Reviewer 1 also raises a similar question and the following is our response.
>
> ***Escaping saddle points faster:***
>
> The escape time $T_{thred}$ of CNC-SGD (Daneshmand et al. 2018), based on Table 3 in their paper, is
> $$T_{thred} = \tilde{O}( \frac{1}{\eta \epsilon}  ),$$
> where $\tilde{O}$ hides the log factor.
> On the other hand, with momentum, we show that (c.f.  (81) and (82) in the appendix )
> $$T_{thred} = \tilde{O}( \frac{(1-\beta)}{\eta \epsilon} ).$$
> We can clearly see that $T_{thred}$ of our result has a dependency $1-\beta$, so it is smaller than that of (Daneshmand et al. 2018) for any same $\eta$, which demonstrates escaping saddle point faster with momentum.
>
> ***Improved iteration complexity of finding a second order stationary point:***
>
> For finding a second order stationary point, our iteration complexity reduces to their results (Theorem 2 in their paper), which is $T=O(\epsilon^{-10})$ to find an $(\epsilon,\epsilon)$ second order stationary point. This is done by setting step size $r$ and $\eta$ appropriately. Specifically, by comparing Table 3 in Appendix E of our paper and Table 3 in their paper, one can see that for the large step size $r$, the small step size $\eta$, the escape time $T_{thred}$, and the total number iterations $T$, they all have the same dependency on failure probability $\delta$, projection lower bound of CNC $\gamma$, the optimality gap $\epsilon$.
>
> However, in order to make sure that we really clarify the concern. Let us be precise. Theorem 2 in (Daneshmand et al. 2018) states that, for CNC-SGD to find an $(\epsilon,\rho^{1/2} \epsilon)$ stationary point, the total number of iterations is $T = O( \frac{ \ell^{10} L^3  } { \delta^4 \gamma^4} \log^2 ( \frac{\ell L}{\epsilon \delta \gamma} ) \epsilon^{-10}  ),$ where $\ell$ is the bound of the stochastic gradient norm $\| g_t \| \leq \ell$ which can be viewed as the counterpart of $c_m$ in our paper. By translating their result for finding an $(\epsilon,\epsilon)$ stationary point, it is $T = O( \frac{ \ell^{10} L^3 \rho^5} { \delta^4 \gamma^4} \log^2 ( \frac{\rho \ell L}{\epsilon \delta \gamma} ) \epsilon^{-10}  )$. On the other hand, using the parameters value on Table~3, we show that $T = 2 T_{thred} ( f(w_0) - \min_w f(w) ) / (\delta  F_{thred}) = O( \frac{ (1-\beta) c_m^9 L^3 \rho^3 (\sigma^2)^3 c_h^2 c'  }{ \delta^4 \gamma^4} \log( \frac{L c_m \sigma^2  c' c_h}{ (1-\beta)  \delta \gamma \epsilon }  )\epsilon^{-10} )$ for Algorithm~2 in Appendix G.
>
> Before making a comparison, we note that their result does not have a dependency on the variance of stochastic gradient (i.e. $\sigma^2$), which is because they assume that the variance is also bounded by the constant $\ell$ (can be seen from (86) in the supplementary of their paper, where the variance terms $\| \zeta_i \|$ are bounded by $\ell$). Following their treatment, if we assume that $\sigma^2 \leq c_m$, then on (71) we can instead replace $(\sigma^2 + 3 c_m^2)$ with $4 c_m^2$ and on (72) it becomes $1 \geq  \frac{ 576 c_m^3 \rho c_r \epsilon }{  (1-\beta)^3  }$. This will remove all the parameters' dependency on $\sigma^2$. Now by comparing $\tilde{O}( (1-\beta) c_m^9  c_h^2 c' \cdot  \frac{ \rho^3 L^3  }{ \delta^4 \gamma^4} \epsilon^{-10} )$ of ours and $T = \tilde{O}( \rho^2 \ell^{10} \cdot \frac{ \rho^3 L^3   }{ \delta^4 \gamma^4}  \epsilon^{-10}  )$ of (Daneshmand et al. 2018), we see that in the high momentum regime where $$(1-\beta) << \frac{ \rho^2 \ell^{10} }{  c_m^9  c_h^2 c' },$$ Algorithm~2 is strictly better than that of (Daneshmand et al. 2018), which means that a higher momentum can help to find a second order stationary point faster. Empirically, we find out that $c' \approx 0$ (Figure 3) and $c_h \approx 0$ (Figure 4) in the phase retrieval problem, so the condition is easily satisfied for a wide range of $\beta$.

---

> > ### Author Response · Authors · 2019-11-11
> > **Response to comment 2~5**
> >
> >
> > === Response to comment 2 ===
> >
> > To address your Question 2, we emphasize that in a non-convex setting one needs to find appropriate conditions under which SGD with momentum converges. The conditions we propose appear to be quite natural, and we show empirically that they hold in at least two settings of interest. We do analyze Algorithm 2, but we note that this is only a slightly-modified version of Algorithm 1, where we occasionally take a larger step. This slight modification is indeed required to obtain the desired guarantee, that Algorithm 2 escapes saddle points and finds a second order stationary point faster with higher momentum parameter $\beta$.
> >
> > === Response to comment 3 ===
> >
> > It seems we were not totally clear in this section of the paper. Let us give a recap to see if this helps elucidate the main points made here. Subsection 3.2.1 is dedicated to showing escaping saddle points faster by stochastic momentum. We begin by stating that the purpose is showing that $T_{thred}$ iterations is enough to escape saddle points. Also, we try to emphasize that the dependency of $T_{thred}$ on the momentum parameter $\beta$ suggests that higher momentum helps speed up the escape from saddle points. Then, we adopt the ``proof by contradiction'' method to bound iteration complexity that follows along the same lines as several related works; that is, we show that a distance lower bound exceeds an associated upper bound. The five lemmas following the first paragraph support the result. For each lemma, we briefly explained what the lemma functions in the proof, either right before or right after the statement. Lemma 1 gives the distance upper bound, Lemma 2,3,4 establish the distance lower bound, and Lemma 5 concludes the contradiction. Lemma 4 is perhaps the most crucial, as it shows that a component of the lower bound is strictly monotone increasing with the momentum parameter $\beta$, which is the key to accelerating the escape.
> >
> > For the notations used in Subsection 3.2.1, they are either introduced before the subsection or highlighted in the subsection (e.g. the $q's$ stuff in Lemma 2), which we also summarize them in the beginning of Appendix D.
> >
> > Please let us know if it is still not clear; we will try hard to have a better presentation.
> >
> > === Response to comment 4 and 5 ===
> >
> > We would be happy to include your version of the update in the final version. But just to be completely clear: your algorithm and the one we proposed are the same, just with different presentation. To see this, let $w_0= w_{-1}$, denote $g_t := \nabla f(w_t, \xi_t)$ the stochastic gradient, and recall that the stochastic momentum $m_t$ is defined as the weighted average of the stochastic gradients $g_t$, which is $m_t:= \sum_{s=0}^t \beta^{t-s} g_s$,
> > we have that
> > $$    w_1 = w_0 - \eta g_0 = w_0 - \eta m_0 $$
> > $$    w_2 = w_1 - \eta g_1 + \beta ( w_1 - w_0 ) = w_1 - \eta g_1 + \beta ( - \eta g_0 ) = w_1 - \eta \sum_{s=0}^{1} \beta^{1-s} g_s  = w_1 - \eta m_1 $$
> > $$ w_3 = w_2 - \eta g_2 + \beta ( w_2 - w_1 )
> > = w_2 - \eta g_2 - \eta \beta g_1 - \eta \beta^2 g_0
> > = w_2 - \eta \sum_{s=0}^{2} \beta^{2-s} g_s  = w_2 - \eta m_2
> > $$
> > Therefore, we do not believe that ``SGD with momentum'' and ``stochastic heavy ball'' refer to different algorithms.
> >
> > We call the popular method ``SGD with stochastic momentum'' sometimes to emphasize that $m_t$ is the weighted average of the stochastic gradients instead of the deterministic gradients. We do not think that our naming is inappropriate, neither do we believe that the name of the popular method has been coined.

---

### Official Review · AnonReviewer1 · 2019-10-23
**Official Blind Review #1**

**Rating:** 3

**Review:**

*Summary*
This paper studies the impact of momentum for escaping Saddle points with SGD (+momentum) in a non-convex optimization setting.
They prove that using a large momentum value (i.e. close to one) provides a better constant for the convergence rate to second order stationary points.
The approach is well motivated by the current seek for a better understanding of the training methods used in practical deep learning. The related work seems to be well addressed.

*Decision*:
I think that this paper is borderline for the following reasons: First I would like to acknowledge the fact that the problem of studying the impact of momentum on SGD is a very challenging problem with a plethora of open questions even in the convex setting. Thus, trying to show evidence of a significant additive value of momentum in the non-convex setting is a very hard problem.
However, I am not fully convinced that this work provides a results that exhibits a regime where momentum helps to escape saddle point faster (or to converge faster).
Thus I am leaning toward a weak reject but I am eager to change my grade if the authors convince me that their results bring theoretical improvements.

*Questions*:
- I would like to be convinced that the analysis use the momentum aspect to improve in some ways the convergence rate of SGD and does not reduce to a perturbation analysis of the SGD method. Because of the constraints on $\beta$ (for instance $1-\beta > L^{-1/3}$) , your $1-\beta$ dependance in the convergence rate cannot be better than $L^{-1/3}$. Thus, one could believe the $\beta$ in the bound is only a hidden proxy for the Lipschitz constant or for $\sigma$.  One way to be convincing about the fact that this bound is interesting would be to provide some regime where we can see improvements compared with related work. (e.g. $L >> \sigma, c’>> 1$) Or maybe if the current theoretical interest of the momentum is only in escaping saddle point better (since its convergence in the convex case is still not well understood) than regular SGD methods.
- To what extend your assumptions are comparable to the ones made in the close related work (papers presented in Table 1) ?



*Minor remarks*:
- You should cite something for the NP-hardness of finding local minima.
- The definition for $(\epsilon,\epsilon)$-second order stationary point would be a bit cleared with two different epsilon $\epsilon_1,\epsilon_2$.
- I think should should put the constraints you have on $\beta$ in the main paper.
-  In definition 2 does $M_t$ depends on $k$ or is it a typo, (the summation index for $G_{s,t}$ depends on $k$ that make the definition confusing) ? Can $G_{s,t}$ be rewritten as $I- \eta (1 - \beta^s)/(1-\beta) \nabla^2f(\omega_t)$ to avoid to introduce an unnecessary sum notation ?
- Usually footnotes are after the punctuation sign (because you want to comment the whole sentence and not its last word).
- Footnote 4 is in the wrong page.

==== After rebuttal ====
I have read the answers of the authors and I still think that their results are too hard to interpret.
I also think that the results are still presented in a misleading way (putting $1-\beta$ in the $O()$ while this quantity is not allowed to vanish since it has to be greater that $L^{-1/3}$).
The interest of the momentum method should be that when using $\beta = 1 - L^{-1/3}$ the escaping time (for instance) is better.
The author did not convince that is was the case.

However I would tend to change my grade to 4 or 5 /10 if it was possible. (i.e. very borderline weak reject)


**Experience Assessment:**

I have read many papers in this area.

**Review Assessment: Checking Correctness Of Derivations And Theory:**

I assessed the sensibility of the derivations and theory.

**Review Assessment: Checking Correctness Of Experiments:**

I assessed the sensibility of the experiments.

**Review Assessment: Thoroughness In Paper Reading:**

I read the paper at least twice and used my best judgement in assessing the paper.

---

> ### Author Response · Authors · 2019-11-11
> **Response to Question 1**
>
> We thank for your valuable comments as we expect these will help shape the paper. We hope the following response addresses your questions.
>
> === Response to Question 1 ===
>
> We appreciate your concern for the relevance of the dependence on $\beta$, particularly in light of the constraints required on this parameter. But would like to emphasize a couple of points:
>
> 1) The $\beta$ dependence on $L$ and $\sigma$ are only artificial. We use these constraints in our proofs but they are mostly artefacts of the analysis. For example, if a function is $L$-smooth, and $L < 1$, then it is also $1$-smooth, so we can assume without loss of generality that $L > 1$. Similarly, the dependence on $\sigma$ is not highly relevant, since we can always increase the variance of the stochastic gradient, for example by adding an $O(1)$ gaussian perturbation.
>
> 2) We would like to emphasize that, even in light of the constraints on $\beta$, our convergence rates are still strictly better (ignoring log terms) than similar works.
>
> To elaborate on this last point, let us give a precise comparison to the result which is closest to ours, (Daneshmand et al. 2018). (Daneshmand et al. 2018) consider a variant of SGD that occasionally uses a large step size and assume CNC holds in stochastic gradient $g_t$. Our analysis framework is built on their work; we consider SGD with momentum that occasionally adopts a large step size and assume CNC holds in stochastic momentum $m_t$ instead.  So Algorithm 2 in our paper reduces to CNC-SGD of (Daneshmand et al. 2018) when $\beta = 0$. Therefore, let us focus on comparing the results of (Daneshmand et al. 2018) for the moment.
>
> ***Escaping saddle points faster:***
>
> The escape time $T_{thred}$ of CNC-SGD (Daneshmand et al. 2018), based on Table 3 in their paper, is
> $$T_{thred} = \tilde{O}( \frac{1}{\eta \epsilon}  ),$$
> where $\tilde{O}$ hides the log factor.
> On the other hand, with momentum, we show that (c.f.  (81) and (82) in the appendix )
> $$T_{thred} = \tilde{O}( \frac{(1-\beta)}{\eta \epsilon} ).$$
> We can clearly see that $T_{thred}$ of our result has a dependency $1-\beta$, so it is smaller than that of (Daneshmand et al. 2018) for any same $\eta$, which demonstrates escaping saddle point faster with momentum.
>
> ***Improved iteration complexity of finding a second order stationary point:***
>
> For finding a second order stationary point, our iteration complexity reduces to their results (Theorem 2 in their paper), which is $T=O(\epsilon^{-10})$ to find an $(\epsilon,\epsilon)$ second order stationary point. This is done by setting step size $r$ and $\eta$ appropriately.
> Specifically, by comparing Table 3 in Appendix E of our paper and Table 3 in their paper, one can see that for the large step size $r$, the small step size $\eta$, the escape time $T_{thred}$, and the total number iterations $T$, they all have the same dependency on failure probability $\delta$, projection lower bound of CNC $\gamma$, the optimality gap $\epsilon$.

---

> > ### Author Response · Authors · 2019-11-11
> > **(continue) Response to Question 1**
> >
> >
> > *** Improved iteration complexity of finding a second order stationary point: ***
> >
> > However, in order to make sure that we really clarify the concern. Let us be precise.
> > Theorem 2 in (Daneshmand et al. 2018) states that, for CNC-SGD to find an $(\epsilon,\rho^{1/2} \epsilon)$ stationary point, the total number of iterations is
> > $T = O( \frac{ \ell^{10} L^3  } { \delta^4 \gamma^4} \log^2 ( \frac{\ell L}{\epsilon \delta \gamma} ) \epsilon^{-10}  ),$ where $\ell$ is the bound of the stochastic gradient norm $\| g_t \| \leq \ell$ which can be viewed as the counterpart of $c_m$ in our paper. By translating their result for finding an $(\epsilon,\epsilon)$ stationary point, it is $T = O( \frac{ \ell^{10} L^3 \rho^5} { \delta^4 \gamma^4} \log^2 ( \frac{\rho \ell L}{\epsilon \delta \gamma} ) \epsilon^{-10}  )$. On the other hand, using the parameters value on Table~3, we show that $T = 2 T_{thred} \big( f(w_0) - \min_w f(w) \big) / (\delta  F_{thred}) = O( \frac{ (1-\beta) c_m^9 L^3 \rho^3 (\sigma^2)^3 c_h^2 c'  }{ \delta^4 \gamma^4} \log( \frac{L c_m \sigma^2  c' c_h}{ (1-\beta)  \delta \gamma \epsilon }  )\epsilon^{-10} )$ for Algorithm~2 in Appendix G.
> >
> > Before making a comparison, we note that their result does not have a dependency on the variance of stochastic gradient (i.e. $\sigma^2$), which is because they assume that the variance is also bounded by the constant $\ell$ (can be seen from (86) in the supplementary of their paper, where the variance terms $\| \zeta_i \|$ are bounded by $\ell$).
> > Following their treatment, if we assume that $\sigma^2 \leq c_m$, then, on (71) in the appendix, we can instead replace $(\sigma^2 + 3 c_m^2)$ with $4 c_m^2$ and (72) becomes $1 \geq  \frac{ 576 c_m^3 \rho c_r \epsilon }{  (1-\beta)^3  }$. This will remove all the parameters' dependency on $\sigma^2$.
> >
> > Now by comparing
> > $\tilde{O}( (1-\beta) c_m^9  c_h^2 c' \cdot  \frac{ \rho^3 L^3  }{ \delta^4 \gamma^4} \epsilon^{-10} )$ of ours and $T = \tilde{O}( \rho^2 \ell^{10} \cdot \frac{ \rho^3 L^3   }{ \delta^4 \gamma^4}  \epsilon^{-10}  )$ of (Daneshmand et al. 2018),
> > we see that in the high momentum regime where $(1-\beta) << \frac{ \rho^2 \ell^{10} }{  c_m^9  c_h^2 c' }$, Algorithm~2 is strictly better than that of (Daneshmand et al. 2018), which means that a higher momentum can help to find a second order stationary point faster. Empirically, we find out that $c' \approx 0$ (Figure 3) and $c_h \approx 0$ (Figure 4) in the phase retrieval problem, so the condition is easily satisfied for a wide range of $\beta$.
> >
> > We appreciate you for raising the question. We have updated the paper accordingly. We add some new remarks right below Theorem 1 to include the discussion and the comparison.

---

> > > ### Author Response · Authors · 2019-11-11
> > > **Response to Question 2 and other remarks**
> > >
> > >
> > > === Response to Question 2 ===
> > >
> > > Q: To what extend your assumptions are comparable to the ones made in the close related work (papers presented in Table 1) ?
> > >
> > > A: To begin, we would like to first emphasize that all the works of escaping saddle points consider some variants of the standard algorithms. They all make some tweaks of the standard algorithms for the ease of analysis. Therefore, it might not be very easy to make a fair comparison, as the assumptions and modifications vary in different works.
> > >
> > > (1) All the works of escaping saddle points make assumptions of Lipschitzness of gradient (smoothness) and Lipschitzness of Hessian. They also make the assumption that the variance of stochastic gradients is bounded. We just follow the standard assumptions. (Daneshmand et al. 2018) and (Staib et al. 2019) also assume that the norm of stochastic gradient $g_t$ is bounded, while we instead assume that the norm of stochastic momentum $m_t$ is bounded. For $\beta=0$, the boundedness of norm becomes equivalent.
> > >
> > > (2)
> > > The design and analysis in all the related works guarantee that there is a sufficiently large projection on the escape direction when the iterate enters the region of saddle points. To achieve this, (Daneshmand et al. 2018) make the CNC assumption so that the stochastic gradient has a sufficiently large projection on the escape direction, while (Staib et al. 2019) implicitly make the CNC assumption instead. For our work, we assume that CNC holds in stochastic momentum $m_t$, which reduces to that of (Daneshmand et al. 2018) when $\beta = 0$. On the other hand, (Ge et al. 2015) and (Jin et al. 2019) consider explicitly adding noise when updating the iterate without making the CNC assumption. We do note that the dependency of iteration complexity on the optimal gap $\epsilon$ is better in (Ge et al. 2015) and (Jin et al. 2019) than the ones in (Daneshmand et al. 2018), (Staib et al. 2019), and ours. Yet, we think that by explicitly introducing the noise, it offers another parameter to leverage (i.e. the variance of noise) and decouples the component on the escape direction from the stochastic gradient itself. The flexibility might be the reason why the works of explicitly adding noise have a better complexity than the works that make the CNC assumption. For the work (Fang et al. 2019), it assumes an inherent property in stochastic gradient which is different from CNC. We conjecture that it is stronger than CNC. Also, the work considers suffix averaging of the iterate while ours does not have the operation.
> > >
> > >
> > > === Response to the minor remarks ===
> > >
> > > We thank you for the thorough reading. We have updated the paper accordingly. For showing that even finding a local minimum can be NP hard, we include the following papers. Among them, (Anandkumar and Ge 2016) [1] show that finding a fourth order stationary point is NP-hard; the paper is also a good pointer for the related works that show some hardness results.
> > >
> > > [1] Efficient approaches for escaping higher order saddle points in non-convex optimization. Anima Anandkumar and Rong Ge, COLT 2016.
> > > [2] Jiawang Nie. The hierarchy of local minimums in polynomial optimization. Mathematical Programming, 2015.
> > > [3] Katta G Murty and Santosh N Kabadi. Some np-complete problems in quadratic and nonlinear programming. Mathematical programming, 1987.
> > > [4] Yurii Nesterov. Squared functional systems and optimization problems. High performance optimization, Springer, 2000.
> > >
> > > --- Regarding $G_{s,t}$---:
> > > Yes, we can write $G_{s,t} := \mathcal{I} - \eta ( 1 - \beta^s)/(1-\beta) \nabla^2 f(w_t)$ to avoid the confusion. Thank you for the suggestion.

---

> > > > ### Comment · AnonReviewer1 · 2019-11-14
> > > > **Artificial dependence on $L$ and $\sigma$**
> > > >
> > > > Thank you these detailed answers.
> > > > Note that I made a typo in my review that I have corrected (the condition on $\beta$ is actually $1-\beta {\color{red}>} L{{\color{red}-}1/3}$)
> > > >
> > > > I do not understand your point that argues that the dependence on  $L$ and $\sigma$ are "not highly relevant".
> > > > If you add Gaussian noise to the gradients then $\sigma$ increases and the convergence is actually slower so the fact that $\sigma$ does appear in the bound do not seem artificial.
> > > >
> > > > Overall in your comparison of rates why don't you just replace $(1-\beta)L^3$ by 1 ?
> > > > Thus your rate would have not dependence in $L$ (why is quite surprising by the way)
> > > >
> > > > Also in you comparison of escape time $T_{thred}$ is there any dependence in $L$ hidden in the $O( \frac{1}{\eta\epsilon})$ ?
> > > >
> > > > I think that the presentation of your results would be more fair and clear if you used the optimal value you manged to get from your theory  ($\beta = 1-L^{-1/3}$). Thus, you could compare your result with (Daneshmand et al. 2018) saying that in a certain regime your rate is better.

---

> > > > > ### Author Response · Authors · 2019-11-14
> > > > > **Response to new comments**
> > > > >
> > > > > We thank you again for giving us a chance of clarifying the concerns.
> > > > >
> > > > > ***Q: I do not understand your point that argues that the dependence on $L$ and $\sigma$ are "not highly relevant". If you add Gaussian noise to the gradients then  increases and the convergence is actually slower so the fact that $\sigma$ does appear in the bound do not seem artificial.
> > > > >
> > > > > ***A: Sorry for the confusion. You are right that the tightest possible constants of $L$ and $\sigma^2$ get worse and the convergence can be slower as the results. We just meant that the constraints of $\beta$ (e.g. $\sigma^2 (1-\beta)^3 > 1$) can be ``artificially'' satisfied by using some loose constants of $L$ or $\sigma^2$. However, the constraints of $\beta$ trivially hold on hard problems (i.e. problems with large smoothness $L$, large variance $\sigma^2$).
> > > > >
> > > > >
> > > > >
> > > > >
> > > > > ***Q: Overall in your comparison of rates why don't you just replace $(1-\beta) L^3$  by $1$ ? Thus your rate would have not dependence in $L$ (why is quite surprising by the way). I think that the presentation of your results would be more fair and clear if you used the optimal value you manged to get from your theory $\beta = 1 - L^{-1/3}$. Thus, you could compare your result with (Daneshmand et al. 2018) saying that in a certain regime your rate is better.
> > > > >
> > > > > ***A: First, we want to point out that there is an $L$ in the $\log$ factor of $T$ so that we cannot totally remove the dependency.
> > > > >
> > > > > In the previous reply, we state that in the high momentum regime where $(1-\beta) << \frac{ \rho^2 \ell^{10} }{  c_m^9  c_h^2 c' }$, using momentum is strictly better than that without using it. That is, any $\beta$ that satisfies the above inequality guarantees our result is better than that of
> > > > > (Daneshmand et al. 2018), including $\beta = 1 - L^{-1/3}$ if it satisfies the constraint.
> > > > >
> > > > > But, perhaps let us use the following way of removing $\beta$ to describe the regime where the acceleration is guaranteed. So the inequality $(1-\beta) << \frac{ \rho^2 \ell^{10} }{  c_m^9  c_h^2 c' }$ gives a lower bound of $\beta$ that guarantees the acceleration, which is $\beta >> \beta_{low}:= 1 -\frac{ \rho^2 \ell^{10} }{  c_m^9  c_h^2 c' }$.
> > > > > On the other hand, based on Remark 1 right below Theorem 1, the upper-bounds of $\beta$ can be re-written as
> > > > > $\beta \leq \beta_{high}:=\min\{ 1 - L^{-1/3}, 1 - (\sigma^2)^{-1/3}, 1 - (c')^{-1/2}\}$. (Let us ignore the constraints $4), 5), 6)$ of $\beta$ by noting that $\eta = O(\epsilon^5)$ and assuming that $\epsilon$ is small.)
> > > > > Hence, there is acceleration for $\beta$ in the range of $[\beta_{low}, \beta_{high}]$, which is non-empty when
> > > > > $$ \beta_{low} < \beta_{high} \iff \max\{ L^{-1/3}, (\sigma^2)^{-1/3}, (c')^{-1/2}\}
> > > > > < \frac{ \rho^2 \ell^{10} }{  c_m^9  c_h^2 c' } .$$
> > > > > This is the regime in which our result is strictly better than that of (Daneshmand et al. 2018). We know that it can be empty (i.e. the regime may not exist). On the other hand, it tells us that when $L$, $\sigma^2$ is large, and/or when the algorithm exhibits nice APCG and GrACE properties such that $c_h$, $c'$ are small (like $c_h \approx 0$, $c' \approx 0$ in phase retrieval), using momentum improves the complexity of (Daneshmand et al. 2018) for finding a second order stationary point.
> > > > >
> > > > > To summarize, our result $T=\tilde{O}( (1-\beta) c_m^9  c_h^2 c' \cdot  \frac{ \rho^3 L^3  }{ \delta^4 \gamma^4} \epsilon^{-10} )$ reduces the complexity of $T = \tilde{O}( \rho^2 \ell^{10} \cdot \frac{ \rho^3 L^3   }{ \delta^4 \gamma^4}  \epsilon^{-10}  )$  (Daneshmand et al. 2018) by a factor of $\tilde{O}( \frac{\rho^2 \ell^{10} }{ (1-\beta) c_m^9  c_h^2 c' })$ with $\beta \in [\beta_{low}, \beta_{high}]$ and the factor is meaningful (i.e. $\geq 1$) in the regime of $$ \max\{ L^{-1/3}, (\sigma^2)^{-1/3}, (c')^{-1/2}\}
> > > > > < \frac{ \rho^2 \ell^{10} }{  c_m^9  c_h^2 c' } .$$
> > > > >
> > > > >
> > > > >
> > > > >
> > > > > ***Q: Also in you comparison of escape time $T_{thred}$ is there any dependence in $L$ hidden in the $\tilde{O}(\frac{1}{\eta \epsilon})$?
> > > > >
> > > > > ***A: First, for finding a second order stationary point, $\eta$ is set appropriately, $\eta = O(\frac{\epsilon^5}{L^2})$. Therefore, the complexity does depend on $L$.
> > > > >
> > > > > However, if the goal is to escape saddle points when using any $\eta$, then the $\log$ factors in the rates of both works do depend on different problem parameters. For the result of (Daneshmand et al. 2018), smoothness $L$ is inside the $\log$ factor. On the other hand, for our result, it is Lipschitzness $\rho$ of the Hessian inside the $\log$ factor. The reason is that we use an inequality of the Lipschitzness of the Hessian for proving Lemma 1, instead of using an inequality of the smoothness as (Daneshmand et al. 2018).

---

### Official Review · AnonReviewer2 · 2019-10-23
**Official Blind Review #2**

**Rating:** 6

**Review:**

This paper makes an interesting theoretical contribution; namely, that SGD with momentum (and with a slight modification to the step-size rule) is guaranteed to quickly converge to a second-order stationary point, implying it quickly escapes saddle points. SGD with momentum is widely used in the practice of deep learning, but a theoretical analysis has remained largely elusive. This paper sheds light theoretical properties justifying its use for deep learning.

Although the paper makes assumptions (e.g., twice differentiable, with smooth Hessian) that are not valid for the most widely-used deep learning models, the theoretical contributions of this paper should nonetheless be of interest to researchers in optimization for machine learning. I recommend it be accepted.

The experiments reported in the paper, including those used to validate the required properties, are for small toy problems. This is reasonable given that the main contribution of the paper is theoretical. However, I would have given a higher rating if some further exploration of the validity of these properties was carried out for problems closer to those of interest to the broader ICLR community. Even if the assumptions aren't satisfied everywhere for typical deep networks, they may be satisfied at most points encountered during training, which would make the contribution even more compelling. This may also help to understand some of the limitations of this analysis.

One other limitation seems to be that Theorem 1 requires using a step size which seems to be much smaller than what one may hope to use in practice. Can you comment on this?



**Experience Assessment:**

I have read many papers in this area.

**Review Assessment: Checking Correctness Of Derivations And Theory:**

I assessed the sensibility of the derivations and theory.

**Review Assessment: Checking Correctness Of Experiments:**

I carefully checked the experiments.

**Review Assessment: Thoroughness In Paper Reading:**

I read the paper at least twice and used my best judgement in assessing the paper.

---

> ### Author Response · Authors · 2019-11-11
> **Response to Reviewer 2**
>
>
> We thank for your valuable comments and suggestions.
>
> === Regarding to the assumptions, specifically, twice differentiable/smooth Hessian ===
>
> Twice differentiable/smooth Hessian are only used for analyzing the process of escaping saddle points. So we conjecture that one can relax the assumptions and introduce the notions like ``locally twice differentiable'' and ``locally smooth Hessian'', meaning that the assumptions only need to hold in the region of the saddle points. Since the gradient norm in the region of the saddle points is small, it implies that the Hessian should not change too much and ``locally smooth Hessian'' should make sense. However, we are not aware of any related works of escaping saddle points introducing any measures of ``locally smooth Hessian''. You might actually point out a good research direction.
>
> === Regarding to the empirical results/experiments ===
>
> We appreciate your acknowledgment of our contributions and pointing out that the properties may only need to be satisfied at some critical points during training deep neural nets. We will keep updating the paper and conducting more thorough experiments.
>
> === Regarding to the small step size ===
>
> We think that it is a gap, for which people in the community haven't have any good remedies yet. Almost all of the theoretical works in nonconvex optimization and deep learning require a small step size (e.g. works of natural tangent kernel, works of showing the global convergence for a two layer neural net). Nevertheless, we want to note that the step size $\eta = O(\epsilon^5)$ in our paper is of the same order as the closely related work (Daneshmand et al. 2018) of escaping saddle points.

---

> > ### Comment · AnonReviewer2 · 2019-11-15
> > **Thanks, I read your responses**
> >
> > Thanks for your responses to my review. I read them and am keeping my review and score as-is.

---

### Decision · Program_Chairs · 2019-12-19

**Decision:**

Accept (Poster)

**Comment:**

This paper studies the impact of using momentum to escape saddle points. They show that a heavy use of momentum improves the convergence rate to second order stationary points. The reviewers agreed that this type of analysis is interesting and helps understand the benefits of this standard method in deep learning. The authors were able to address most of the concerns of the reviewers during rebutal, but is borderline due to lingering concerns about the presentation of the results. We encourage the authors to give more thought to the presentation before publication.